# Requirement of Smurf-mediated endocytosis of Patched1 in sonic hedgehog signal reception

Shen Yue[1], Liu-Ya Tang[2†], Ying Tang[1†], Yi Tang[2†], Qiu-Hong Shen[1†], Jie Ding[1], Yan Chen[1], Zengdi Zhang[1], Ting-Ting Yu[1], Ying E Zhang[2*], Steven Y Cheng[1*]

[1]Department of Developmental Genetics, School of Basic Medical Sciences, Nanjing Medical University, Nanjing, China; [2]Laboratory of Cellular and Molecular Biology, Center for Cancer Research, National Cancer Institute, Bethesda, United States

**Abstract** Cell surface reception of Sonic hedgehog (Shh) must ensure that the graded morphogenic signal is interpreted accordingly in neighboring cells to specify tissue patterns during development. Here, we report endocytic sorting signals for the receptor Patched1 (Ptch1), comprising two 'PPXY' motifs, that direct it to degradation in lysosomes. These signals are recognized by two HECT-domain ubiquitin E3 ligases, Smurf1 and Smurf2, which are induced by Shh and become enriched in Caveolin-1 lipid rafts in association with Ptch1. Smurf-mediated endocytic turnover of Ptch1 is essential for its clearance from the primary cilium and pathway activation. Removal of both Smurfs completely abolishes the ability of Shh to sustain the proliferation of postnatal granule cell precursors in the cerebellum. These findings reveal a novel step in the Shh pathway activation as part of the Ptch1 negative feedback loop that precisely controls the signaling output in response to Shh gradient signal.

**\*For correspondence:**
sycheng@njmu.edu.cn (SYC);
zhangyin@mail.nih.gov (YEZ)

†These authors contributed equally to this work

**Competing interests:** The authors declare that no competing interests exist.

**Reviewing editor**: Robb Krumlauf, Stowers Institute for Medical Research, United States

## Introduction

The secreted Sonic hedgehog (Shh) protein specifies spatial tissue patterns during development by providing positional cues embedded in its concentration gradient (*Jiang and Hui, 2008*; *Robbins et al., 2012*; *Ryan and Chiang, 2012*). During embryogenesis, neighboring progenitor cells in a developing field are able to discern incremental changes in the Shh signal strength and adopt their respective fate accordingly (*Ribes and Briscoe, 2009*; *Balaskas et al., 2012*). This ability requires a cell surface reception system that can transform the graded Shh signal into different levels of signaling output, but how this is accomplished is poorly understood. In the adult, Shh plays a crucial role in guiding the differentiation of tissue-specific stem cells (*Jaks et al., 2008*; *Shin et al., 2011*; *Arwert et al., 2012*), and inappropriate activation of Shh signaling could be the culprit that underlines neoplastic growth in the gut epithelium (*Nielsen et al., 2004*) or lead to outright cancers (*Scales and de Sauvage, 2009*; *Stecca and Ruiz, 2010*; *Northcott et al., 2012*).

At the cell surface, whereas a network of membrane proteins, including Hip1 (*Chuang et al., 2003*), Gas1 (*Lee et al., 2001*), Boc/iHog, and Cdo/Boi (*Okada et al., 2006*; *Tenzen et al., 2006*; *Yao et al., 2006*; *Beachy et al., 2010*), bind Shh and control the range and competence of its receiving cells, the core of Shh signal reception consists of Patched1 (Ptch1), a 12-pass membrane receptor that acts negatively on Smoothened (Smo), a G-protein-coupled, receptor-like signal transducer (*Rohatgi and Scott, 2007b*). Binding of Shh to Ptch1 alleviates the Ptch1 inhibition of Smo, allowing the signal to propagate to three Gli proteins, the transcriptional effectors of the pathway, and activate the expression of target genes, including pathway components Ptch1 and Gli1 themselves. Since Gli1 is a potent activator of Shh target genes, its induction by the ligand ensures that pathway activation will attain the

**eLife digest** Sonic hedgehog protein fulfils many vital roles in establishing the body plan of multicellular organisms during development. And in adult organisms it regulates the stem cells that maintain organs and tissues. In the embryo, Sonic hedgehog is secreted by certain cells to create a concentration gradient; cells then measure this concentration to work out where they are, which allows them to develop into the right sort of cells. However, many details of this process are not completely understood.

At the core of this process are the interactions between the Sonic hedgehog protein, a receptor called Patched1 that is found on plasma membranes, and another membrane protein called Smoothened. The job of Smoothened is to activate proteins that enter the cell nucleus and 'switch on' the pathway's target genes, which encode Patched1 and a number of other proteins. The role of Patched1, on the other hand, is to repress Smoothened. However, when sonic hedgehog binds to Patched1, the latter is unable to repress Smoothened.

Increasing the production of Patched1 is thought to serve two main roles: it prevents activation of the Sonic hedgehog pathway, and it prevents the Sonic hedgehog protein spreading to neighboring cells (by binding to it). But how is the level of Patched1 itself regulated? Yue et al. now report that two proteins, called Smurf1 and Smurf2, perform this regulation role in mammalian cells.

Smurf1 and Smurf2 are enzymes that attach a molecule called ubiquitin to proteins, setting in train a series of events that leads to the degradation of the protein. Yue et al. now show that Smurf1 and Smurf2 recognize a signal on Patched1 and perform a similar modification, causing the Patched1 to be internalized through an alternate pathway and degraded in lysosomes. This series of events ultimately allow the Sonic hedgehog pathway to be activated.

The work of Yue et al. exposes a critical enzymatic step that sorts unbound Patched1 receptors from those that are bound to Sonic hedgehog proteins. Further research is needed to determine if this signaling pathway can be manipulated for therapeutic purposes.

intended effect in a positive feedback loop. On the other hand, induction of the inhibitory Ptch1 amounts to a negative feedback control, which was regarded crucial to the interpretation of the Shh gradient signal (*Ribes and Briscoe, 2009*). In effect, Ptch1 serves two roles in Shh signaling: first, it acts cell autonomously in suppressing the downstream pathway, and second, the excessive Ptch1 induced by Shh acts as a sink in limiting the spread of the ligand, thereby affecting the neighboring cells in a non-cell autonomous fashion (*Chen and Struhl, 1996*; *Torroja et al., 2004*). However, it is not clear what counteracts the induction of Ptch1 to achieve the precision of the regulation.

For many years, Ptch1 and Smo have been seen in punctate intracellular vesicles in both *Drosophila* and mammalian cells (*Capdevila et al., 1994*; *Ramirez-Weber et al., 2000*; *Zhu et al., 2003*; *Li et al., 2012*), and their trafficking between the cytoplasmic membrane and intracellular vesicles found to be crucial to the activation of the Hedgehog pathway (*Denef et al., 2000*; *Incardona et al., 2000*; *Zhu et al., 2003*; *Nakano et al., 2004*; *Lu et al., 2006*; *Milenkovic et al., 2009*; *Li et al., 2012*). It is known that ligand engagement of *Drosophila* receptor Ptc triggers its internalization and membrane presentation of Smo, but membrane trafficking of Ptch1 and Smo in mammalian cells has an added complexity in that Shh signals through the primary cilium (*Huangfu et al., 2003*; *Corbit et al., 2005*; *Goetz and Anderson, 2009*), a microtubule-based membrane protrusion that emanates from the interphase centrioles (*Lefebvre and Rosenbaum, 1986*; *Pazour and Witman, 2003*; *Nachury et al., 2010*). The prevailing model for mammalian Shh activation entails Ptch1 exiting from and Smo translocating into the primary cilium (*Rohatgi et al., 2007a*; *Kovacs et al., 2008*). Some data suggest that Smo trafficking through membranous compartments is controlled by small lipids and the sterol-sensing domain of Ptch1 (*Martin et al., 2001*; *Bijlsma et al., 2006*; *Corcoran and Scott, 2006*; *Yavari et al., 2010*). Since the structural framework of Ptch1 resembles that of bacterial amino acid transporters (*Carstea et al., 1997*), it is conceivable that Ptch1 controls Smo activity or trafficking through such a small molecular intermediate. However, little evidence is available to account for how Ptch1 internalization through endocytosis is regulated, and it is unclear whether ciliary trafficking and endocytosis are obligatorily coupled (*Nachury et al., 2010*).

Receptor endocytosis plays crucial roles in coordinating the strength and duration of many cell signaling systems (*Piddini and Vincent, 2003*; *Polo and Di Fiore, 2006*). At various steps of the endocytic pathway, from the plasma membrane to the endosomes, receptors can be sorted to the proteolytic lumens of lysosomes, leading to desensitization, or back to the plasma membrane for a rapid recovery of cellular responsiveness. In addition to the classical Clathrin-mediated endocytosis, recent advances indicate that membrane receptors are also internalized through lipid rafts (*Le Roy and Wrana, 2005*; *Lajoie and Nabi, 2010*), which are specialized membrane domains enriched in cholesterol and sphingomyelin and stabilized by Caveolin 1 (Cav-1) (*Allen et al., 2007*). Unlike the Clathrin-mediated endocytosis, cargos of caveolae were shown to be unloaded to late endosomes, thereby bypassing early endosomes (*Quirin et al., 2008*; *Hayer et al., 2010*; *Sandvig et al., 2011*). A major forward endocytic sorting signal is ubiquitination (*Hicke and Dunn, 2003*; *Mukhopadhyay and Riezman, 2007*; *Hayer et al., 2010*), and many HECT-domain E3 ligases have been implicated in the Ubiquitin control of endocytosis, including Smurf2 (*Di Guglielmo et al., 2003*; *Metzger et al., 2012*), which was first identified as a negative regulator of TGF-β/BMP signaling (*Kavsak et al., 2000*; *Zhang et al., 2001*). Here, we present evidence that Smurf1 and Smurf2 are the Ubiquitin E3 ligases that promote Ptch1 movement from lipid rafts to late endosomes for subsequent degradation in lysosomes. This movement is essential for Ptch1's clearance from primary cilia, Shh pathway activation, and the role of Shh in sustaining the proliferation of cerebellar granule cell precursors. In light of the negative feedback control of Shh signaling by Ptch1, this destruction system would allow the level of signaling output to be set precisely according to the level of the Ptch1 protein.

## Results

### Both PPXY-motifs deletion and endocytosis blockade cause Ptch1 to accumulate in lipid rafts

The C-terminal tails of *Drosophila* Ptc and mouse Ptch1 play an important role in determining its membrane distribution and stability, possibly through the highly conserved 'PPXY' motif (*Lu et al., 2006*; *Kawamura et al., 2008*), which is recognized by the WW domain frequently found in HECT-domain E3 ligases (*Metzger et al., 2012*). Mammalian Ptch1 contains an evolutionarily conserved C-terminal 'PPXY' motif and a second one in the third intracellular loop (*Figure 1—figure supplement 1*), whereas Ptch2 does not and is quite stable (*Kawamura et al., 2008*). Under a confocal microscope and in transfected murine embryonic fibroblasts (MEFs), Ptch1-GFP was primarily detected in punctate vesicles (*Figure 1A*), consistent with what was reported in COS and HeLa cells (*Incardona et al., 2000*; *Karpen et al., 2001*); a large proportion of these Ptch1-GFP vesicles were likely to be endosomes (see below and in *Martin et al., 2001*; *Incardona et al., 2002*). In light of the ubiquitination control of endocytosis, we suspected that the 'PPXY' motifs of Ptch1 might be the signal that regulates its turnover through endosomes and lysosomes. To test this hypothesis, we sought to determine how Ptch1 engages the endocytic pathway by focusing our attention at the rim of the plasma membrane, where treatment with conditioned medium (CM) from HEK293 cells expressing the N-terminal signaling fragment of Shh (ShhN) for 1 hr rendered some of the Ptch1-GFP vesicles also positive for Cav-1 (*Figure 1A*), a specific marker of lipid rafts. To quantify the colocalization, we sampled 10 randomly selected rim areas from different cells imaged for each data point and calculated the colocalization coefficient. The results indicated that ShhN almost doubled the colocalization coefficient between Ptch1-GFP and endogenous Cav-1 from 0.37 ± 0.04 to 0.68 ± 0.04 (*Figure 1A,B*). Blocking late endosome/lysosome passage with chloroquine (Chlq) and concanavalin A (ConA) or lysosomal proteolysis with leupeptin (Leu) showed similar effects (*Figure 1A,B*, *Figure 1—figure supplement 2*). In contrast, the mutant Ptch1Δ2PY-GFP that lacks both 'PPXY' motifs exhibited a higher level of colocalization with Cav-1 than its wildtype counterpart even without ShhN treatment (*Figure 1A,B*). Some of the Ptch1-GFP vesicles at the plasma rim were also positive for Clathrin heavy chain that marks the Clathrin-coated pits, but in contrast to the ligand-inducible enrichment in Cav-1 lipid rafts, the fraction of Ptch1-GFP in Clathrin-coated pits was affected neither by ligand treatment nor deletion of the 'PPXY' motifs (*Figure 1C,D*). To complement the confocal imaging experiments, we conducted a co-sedimentation experiment in a discontinuous sucrose density gradient and found that both Ptch1-FLAG and Ptch1Δ2PY-FLAG co-sedimented with Cav-1 in 20% and 25% buoyancy fractions (*Figure 1E*), indicating that when expressed exogenously, Ptch1-FLAG can find its way into Cav-1 positive lipid rafts even without Shh induction. Thus, both deletion of the 'PPXY' motifs and blocking endocytosis cause Ptch1 to accumulate in lipid rafts.

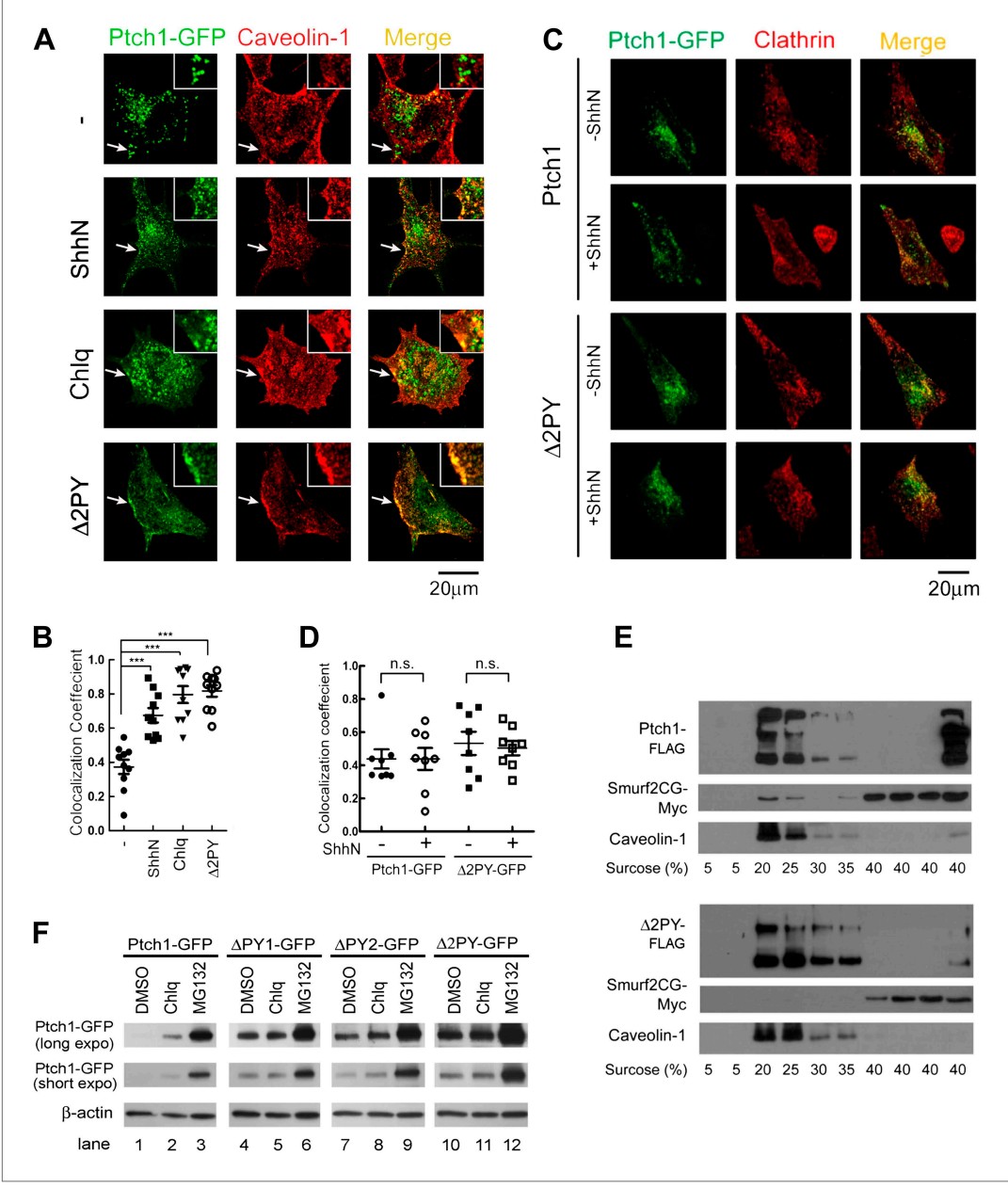

**Figure 1**. The PPXY motifs define sorting signals from lipid rafts to late endosomes. (**A**) Confocal images showing colocalization of exogenously expressed Ptch1-GFP or Δ2PY (green) with native Cav-1 (red) at the rim of the plasma membrane, and (**B**) calculation of the colocalization coefficients in (**A**) in transfected MEFs. ShhN-CM and Chlq were added 1 hr prior to fixation. The chamber slides were chilled at 4°C for 20 min and then shifted to 37°C for another 20 min before fixation with 4% paraformaldehyde in PBS. The cells were then permeabilized with 0.5% Triton X-100 and stained with an antibody for Cav-1. (**C**) Representative images and (**D**) quantification of colocalization between Ptch1-GFP and clathrin heavy chain. MEFs were transfected with Ptch1-GFP or PtchΔ2PY-GFP, then treated with ShhN or Ctrl medium for 1 hr before fixation. (**E**) Western analyses of sucrose gradient fractions showing Ptch1-FLAG co-sedimented with Smurf2CG-Myc and Cav-1. Δ2PY was inefficient in bringing Smurf2CG-Myc into Cav-1 positive sedimentation fractions. (**F**) Western blot analyses of stabilities of Ptch1 and the 'PPXY' motif mutants in MEFs. Chlq or MG132 treatment was carried out for 4 hr. The confocal images were taken with a 63x objective, and the insets in 1A were digitally magnified. Bars represent mean ± standard deviation (SD). Statistical analyses were performed by two-tail Student's $t$ test. ***p<0.001, and *n.s.*, not statistically significant (p>0.05).

*Figure 1. Continued on next page*

*Figure 1. Continued*
The following figure supplements are available for figure 1:
**Figure supplement 1**. Position and sequence alignment of 'PPXY' motifs.
**Figure supplement 2**. Lysosomal inhibitors cause Ptch1-GFP to accumulate in lipid rafts.

Since an end point of endocytosis is degradation in lysosomes, we further asked if wildtype Ptch1 and 'PPXY' motif mutants accumulate differently in the presence of proteasomal or lysosomal blocker. When expressed in MEFs, Ptch1-GFP was an unstable protein; the bulk of which appeared to turnover via proteasomes as Ptch1-GFP accumulated to a very high level in the presence of MG132 (*Figure 1F*, compare lanes 1 and 3). A small portion of Ptch1-FLAG appeared to turnover via lysosomes as indicated by the moderate level of accumulation in the presence of lysosomal inhibitor Chlq (*Figure 1F*, lanes 1 and 2). In contrast, Ptch1 mutants lacking either one of or both 'PPXY' motifs were relatively stable when expressed in MEFs, although they could be induced to accumulate further by MG132 but not Chlq (*Figure 1F*, lanes 4–12). These results suggest that Shh promotes turnover of at least a portion of ectopically expressed Ptch1 via endosomes and lysosomes, but the entry point is likely the Cav-1 positive lipid rafts rather than the conventional clathrin-coated pits.

## The 'PPXY' motifs define an endocytic sorting signals of Ptch1

To ascertain if the 'PPXY' motifs are the actual signal that sorts Ptch1 from lipid rafts to endosomes/lysosomes, we asked if Ptch1-GFP or Δ2PY-GFP could be identified in early endosomes, late endosomes, or lysosomes, which are marked by Rab5-RFP, Rab7-RFP, or Lamp1-RFP, respectively. In the absence of ShhN, Ptch1-GFP and Rab7-RFP could be readily detected together in punctate vesicles, and ShhN treatment drastically increased that colocalization as indicated by colocalization coefficient, which increased from 0.29 ± 0.03 to 0.51 ± 0.02 (*Figure 2A,B*). Similar colocalization between Ptch1-GFP and endogenous Rab7 was also observed under ShhN treatment using specific antibodies (*Figure 2—figure supplement 1*). We could not detect vesicles marked positively with both Ptch1-GFP and Lamp1-RFP or Ptch1-GFP and endogenous Lamp1-RFP without blocking lysosomal enzymes by leupeptin (*Figure 2—figure supplement 2A*), but colozalization between Ptch1-GFP and endogenous Lamp1 was revealed with the use of leupeptin (*Figure 2C*). We did not see Ptch1-GFP colocalizing with either transfected Rab5-RFP (*Figure 2—figure supplement 2B*) or endogenous Rab5 (*Figure 2—figure supplement 3*) without or with ShhN treatment. These observations are consistent with the notion that endocytic cargos of caveolae are unloaded directly to late endosomes, bypassing early endosomes (*Quirin et al., 2008*; *Hayer et al., 2010*; *Sandvig et al., 2011*). In contrast to Ptch1-GFP, Δ2PY-GFP was never found together with any of the three endosome/lysosome markers and ShhN treatment caused no statistically significant change thereof (*Figure 2A–C*, *Figure 2—figure supplements 1 and 2*), indicating that Shh is not able to induce Δ2PY to move beyond lipid rafts to enter late endosomes.

The current paradigm stipulates that Shh induces Ptch1 exit from the primary cilium during signaling (*Rohatgi et al., 2007a*; *Kovacs et al., 2008*). This prompted us to ask if ciliary export or its structural integrity is prerequisite to endocytosis of Ptch1 by comparing the abilities of Ptch1-GFP to associate with Rab7-RFP in *Kif3a*$^{−/−}$ or otherwise isogenic control MEFs. Although *Kif3a*$^{−/−}$ MEFs do not make cilia (*Chen et al., 2011*), Ptch1-GFP could still proceed to late endosomes/lysosomes under the influence of ShhN unabatedly (*Figure 2D,E*), implying that Ptch1 endocytosis is downstream from or independent of ciliary trafficking.

Based on results from the above several lines of investigation, we conclude that the 'PPXY' motifs constitute sorting signals that direct Ptch1 to move into late endosomes for turnover in lysosomes. This sorting event likely takes place in Cav-1 positive lipid rafts since Δ2PY accumulates there in the absence of this signal.

## Ptch1 endocytosis is required for the activation of Shh signaling

Ptc or Ptch1 endocytosis has been observed in cells from *Drosophila* to mammals for some time (*Denef et al., 2000*; *Incardona et al., 2000, 2002*; *Martin et al., 2001*; *Torroja et al., 2004*; *Lu et al., 2006*), but its role was primarily attributed to ligand sequestration or clearing (*Incardona et al., 2000*; *Torroja et al., 2004*). In *Drosophila*, the role of Ptc in ligand sequestration has been shown to be

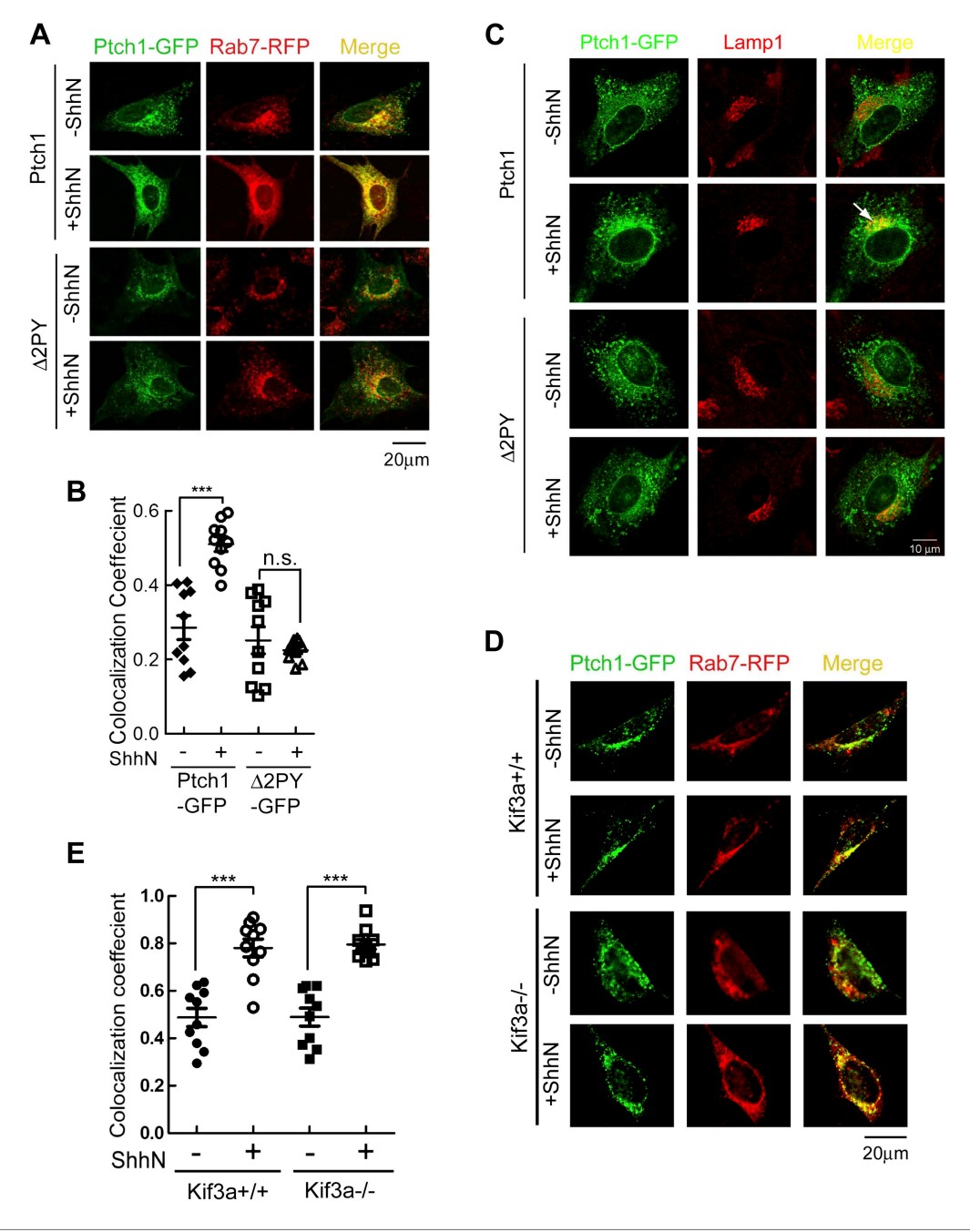

**Figure 2**. PPXY motifs are required for Shh-induced endocytosis of Ptch1. (**A**) Confocal images showing colocalization of Ptch1-GFP or Δ2PY (green) with Rab7-RFP (red), and (**B**) calculation of the colocalization coefficients in (**A**) in transfected MEFs. (**C**) Confocal images showing localization of Ptch1-GFP or Δ2PY (green) in vesicles marked anti-Lamp1 (red) in the presence of 1 mg/ml leupeptin. (**D**) Confocal imaging and (**E**) calculation of colocalization coefficient of Ptch1-GFP and Rab7-RFP in Kif3a$^{-/-}$ and control MEFs. ShhN treatment was for 1 hr and the cells were processed as in *Figure 1A*. Statistical analyses were performed by two-tail Student's *t* test. ***p<0.001, and *n.s.*, not statistically significant (p>0.05).

The following figure supplements are available for figure 2:

**Figure supplement 1**. Shh promotes colocalizaiton of Ptch1-GFP with endogenous Rab7 in late endosomes.

*Figure 2. Continued on next page*

*Figure 2. Continued*

**Figure supplement 2**. Lack of colocalization of Ptch1-GFP or Δ2PY-GFP with exogenous Rab5-RFP and Lamp1-RFP without blocking lysosomal turnover.

**Figure supplement 3**. Ptch1-GFP or Δ2PY-GFP was not found in early endosomes marked by anti-Rab5 immuno-fluorescence staining.

separable from that of signaling based on analyses of certain mutants (*Chen and Struhl, 1996*; *Torroja et al., 2004*). However, we observed that when re-expressed in *Ptch1⁻/⁻* MEFs, the 'PPXY' motif mutants accumulated in the primary cilium, in contrast to their wildtype counterpart; ShhN treatment effectively forced Ptch1-GFP to exit the primary cilium, but it was less effective against these mutants (*Figure 3A,B*). Ciliary accumulation of the 'PPXY' motif mutants is likely a consequence of their inability to endocytose, rather than a specific defect of ciliary export, since these mutants also accumulate in lipid rafts (*Figure 1A,B*) and blocking endocytosis with high concentration of leupeptin showed a similar effect without or with ShhN treatment (*Figure 3—figure supplement 1*). Combined with results from the stability experiment (*Figure 1F*), this observation indicated that these two 'PPXY' motifs play an equivalent role in regulating Ptch1 function in cilia. To support this notion, we made temporal measurements of endogenous Smo translocating into the primary cilium, which is an obligatory early event of Shh signaling and was reported as concurrent to the exit of Ptch1 therefrom (*Rohatgi et al., 2007a*). In *Ptch1⁻/⁻* MEFs, immunofluorescence staining showed that Smo was constitutively present in the primary cilium (*Figure 3C*), as expected (*Corbit et al., 2005*; *Rohatgi et al., 2007a*; *Kovacs et al., 2008*). Re-introducing Ptch1-GFP cleared Smo out of the primary cilium, but ShhN treatment allowed Smo to move back in to nearly its full extent within 4 hr (*Figure 3C,D*). Conversely, ShhN treatment triggered the ciliary export of Ptch1 at a rate comparable to that of Smo import (*Figure 3C*, and compare *Figure 3D,E*). Re-introducing Δ2PY, on the other hand, only allowed a substantially lower level of Smo to be imported back into cilia after ShhN treatment and Δ2PY was itself resistant to Shh-induced export (*Figure 3C,E*).

As a ligand-binding and inhibitory receptor, the functions of *Drosophila* Ptc are twofold; first acting through Smo, Ptc negatively regulates downstream pathway signaling cell autonomously, and second, through ligand sequestration Ptc suppresses Hh signaling in neighboring cells. To determine if Ptch1 endocytosis impinges on downstream pathway activation, we measured the ability of Ptch1-GFP or Δ2PY to confer Shh inducibility to the 8xGliBS-luc reporter in *Ptch1⁻/⁻* MEFs. When co-transfected with Ptch1-GFP, the 8xGliBS-luc reporter showed a robust inductive response to ShhN, resulting in a dose–response curve typical of a substrate-enzyme relationship; however, this reporter was barely induced by ShhN when it was co-transfected with Δ2PY (*Figure 4A*). The Shh signaling blockade imposed by Δ2PY could be by-passed by siRNA-mediated knockdown of Sufu (*Figure 4B*), a downstream negative regulator, suggesting that the blockade is pathway-specific and occurs upstream of Sufu function. So far all our evidence points to inability of the 'PPXY' motif mutants to undergo Shh-induced endocytosis rather than a defect in their intrinsic activity. Indeed, in *Ptch1⁻/⁻* MEFs, these mutants were equally effective as wildtype Ptch1 or cyclopamine in suppressing 8xGliBS-luc reporter independent of the Shh ligand (*Figure 4C*). Finally to address the effect of the 'PPXY' motifs deletion on the non-cell-autonomous function of Ptch1, we designed a 'mixing' experiment, in which *Ptch1⁻/⁻* MEFs re-expressing wildtype Ptch1-GFP or Δ2PY-GFP were mixed at 5 to 1 ratio with a line of stable NIH3T3 cells harboring the genomically integrated 8xGliBS-luc reporter (*Chen et al., 2011*). In the presence of limiting amount of ShhN (1:64 dilution of the conditioned medium), Δ2PY showed a robust inhibition of the ligand-induced reporter activity in the neighboring cells; however this effect was nullified at high ShhN concentration (1:16 dilution) (*Figure 4D*).

In summary, our data indicate that whereas Ptch1 engagement to the ligand may have a nominal effect of internalizing Shh, it can be also regarded as an interaction that allows Shh to induce Ptch1 clearance from the primary cilium, the site of Shh signaling, and this regulation equally impinges on both cell and non-cell autonomous signaling functions of Ptch1.

## Smurf1 and Smurf2 are E3 ligases required for Shh signaling

Previously, the C-terminal domain of *Drosophila* Ptc was shown to be recognized by Nedd4 HECT-domain E3 ligase (*Lu et al., 2006*). We expressed mouse Nedd4 and Nedd4l together with Ptch1-FLAG

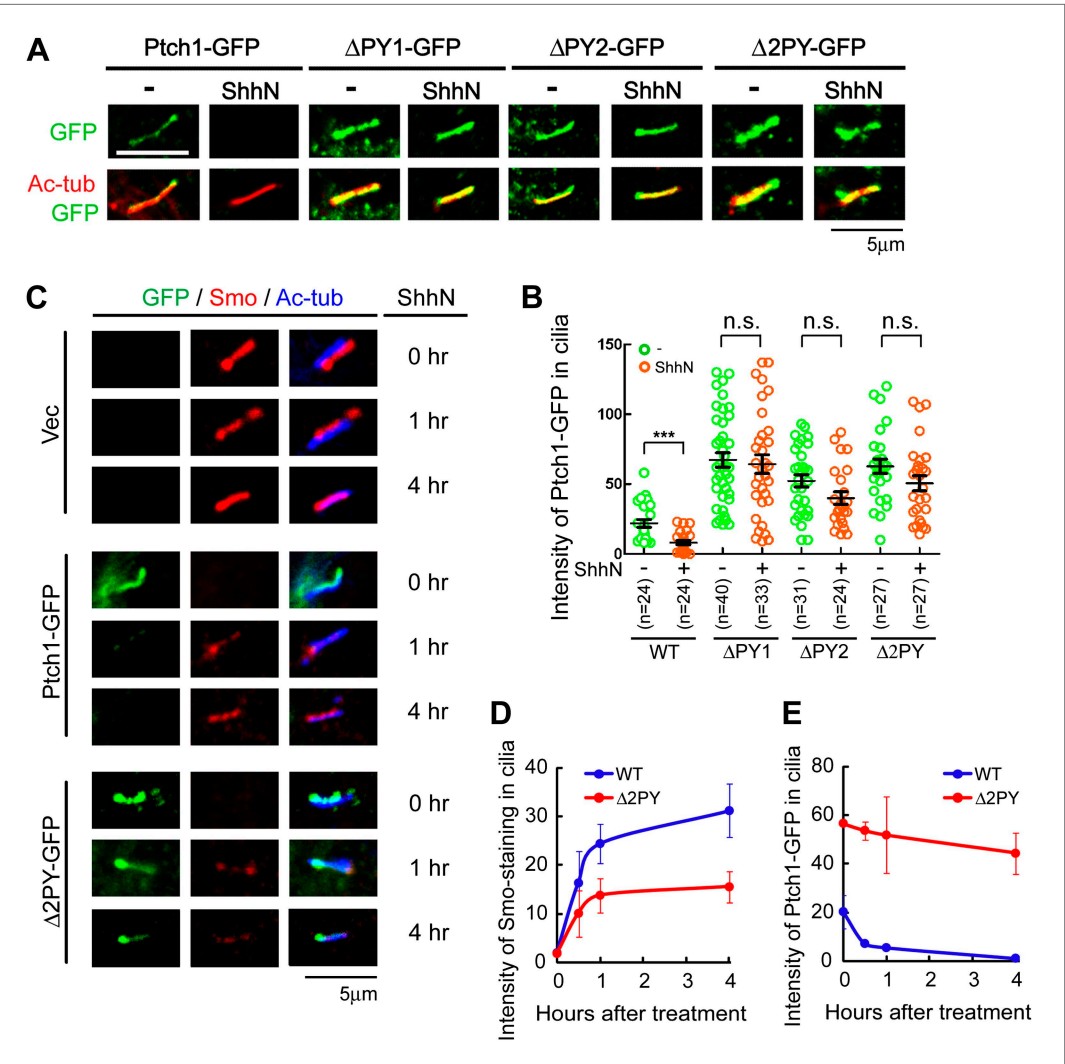

**Figure 3**. The 'PPXY' motifs regulate the opposing movements of Ptch1 out of and Smo into the primary cilium. (**A**) Representative confocal images and (**B**) distribution of GFP fluorescence showing accumulation of the 'PPXY' motif mutants of Ptch1 in primary cilia in the absence or presence of ShhN. Two-tail Student's *t* test was used for statistical analysis. ***p<0.001, n.s., not significant (p>0.05). (**C**) Immunofluorescence of GFP as well as endogenous Smo (red) and acetylated tubulin (blue) staining in Ptch1$^{-/-}$ MEFs transfected with Ptch1-GFP or Δ2PY. (**D**) Quantification of anti-Smo staining and (**E**) GFP fluorescence as in (**C**). Only transfected GFP positive cells were counted for the ciliary localization of endogenous Smo. In all of the above experiments, transfected cells were grown to confluence and then serum-starved for 24 hr to allow for ciliogenesis. ShhN-CM treatment was for 24 hr, or as indicated.

The following figure supplements are available for figure 3:

**Figure supplement 1**. Inhibition of Lysosomal turnover dampens Shh-induced ciliary exit of Ptch1-GFP.

in HEK293 cells, and found that neither one promoted Ptch1 degradation, and several other HECT-domain E3 ligases including Wwp1, Wwp2, Huwe1, Herc1, Herc3, Herc4, Herc6, Hecw1, and Hecw2 also showed no effect, but co-expression of Smurf1 or Smurf2 did (**Figure 5A,B**). Consistent with a specific role, the ligase deficient Smurf1CA and Smurf2CG mutants failed to influence Ptch1 stability (**Figure 5B**). Since the 'PPXY' motif mutants accumulated in cilia, we asked if knockdown of either Smurf or both with siRNAs could augment the ciliary localization of Ptch1-GFP. We found this was the case in NIH3T3 cells without (**Figure 5C,D**) or with ShhN treatment (**Figure 5—figure supplement 1**). Because Smurf2 is known to direct the TGF-β type I receptor and the μ opioid neuropeptide receptor

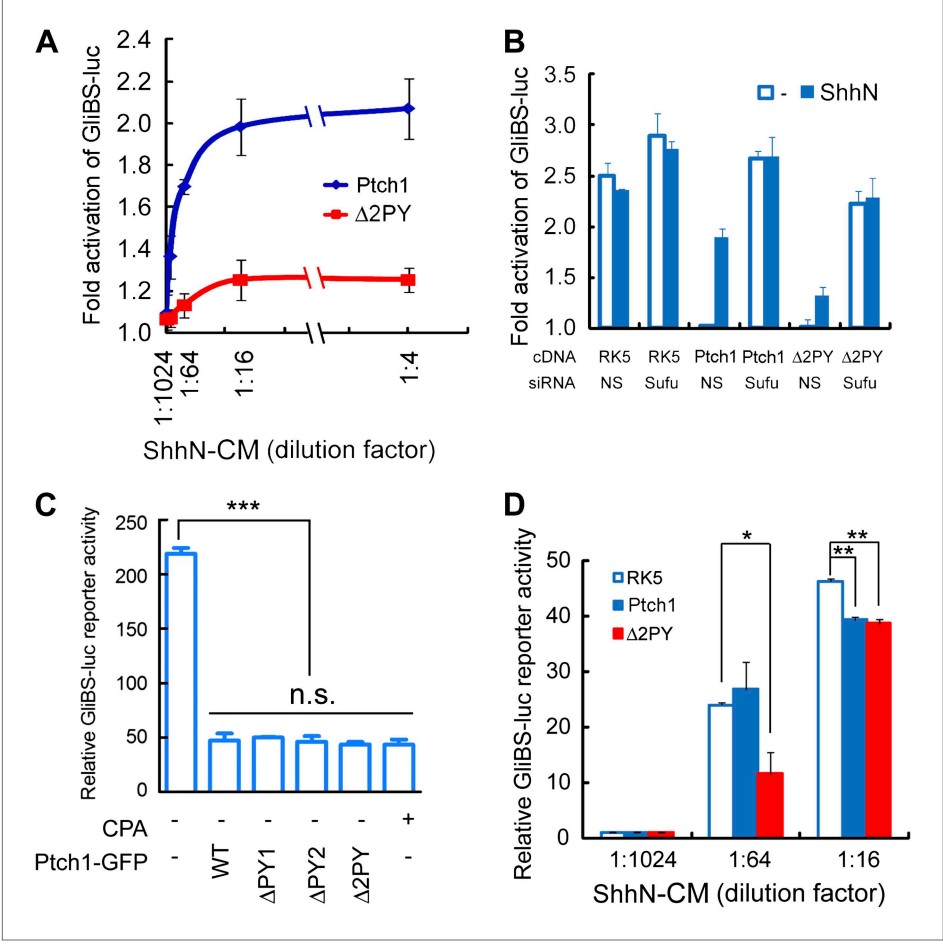

**Figure 4**. The 'PPXY' motifs are required for eliciting both cell and non-cell autonomous transcriptional responses to Shh. (**A**) Luciferase assays for Ptch1 and the Δ2PY mutant in Ptch1⁻/⁻ MEFs that were transfected together with the 8xGliBS-luc reporter construct. Each data point was obtained in triplicate and the error bars denote the standard error. (**B**) Rescuing Shh induction blockade imposed by Δ2PY using siSufu in Ptch1⁻/⁻ MEFs. The experiment was set up as in (**A**) except that Sufu was knocked down by siRNA at the same time as cDNA transfection and 1:16 dilution of ShhN-CM was used. (**C**) Relative activities of the GliBS-luc reporter that was co-expressed with Ptch1 or the ΔPY mutants in Ptch1⁻/⁻ MEFs without ShhN-CM treatment. The ΔPY mutants displayed same inhibitory effect as WT Ptch1. (**D**) Non-cell autonomous inhibition of GliBS-luc reporter in neighboring cells. Ptch1⁻/⁻ MEFs transfected with Ptch1-FLAG, Δ2PY, or the vector control were mixed at 5:1 ratio with NIH3T3:GliBS-luc reporter cells. The cells were given ShhN-CM for 24 hr, and two-tail Student's t test was used for statistical analyses. *p<0.05, **p<0.01, ***p<0.001, and n.s., not significant (p>0.05).

to endocytic turnover (**Di Guglielmo et al., 2003**; **Henry et al., 2012**), we posited that Smurf1 and Smurf2 might be the enzymes that control Ptch1 endocytosis and chose them for further analysis.

Smurf1 and Smurf2 share redundant functions during development as individually knockout Smurf1⁻/⁻ or Smurf2⁻/⁻ mice are healthy and fertile, but the embryos lacking both genes were not able to develop to term (**Yamashita et al., 2005**; **Narimatsu et al., 2009**; **Blank et al., 2012**). To assess the role of Smurfs in Shh signaling, we quantified the transcriptional responses of endogenous Gli1 by RT-PCR (**Figure 5E**) and RT-qPCR (**Figure 5F**) in MEFs with different Smurf genetic background, and found that silencing Smurf1 and Smurf2 simultaneously in wildtype MEFs completely abolished Shh induction of Gli1 (**Figure 5E,F**). MEFs that lack one of the two Smurf genes still mounted a considerable Gli1 transcriptional response to ShhN; however, silencing the remaining Smurf2 allele in Smurf1⁻/⁻ or Smurf1 allele in Smurf2⁻/⁻ MEFs, respectively, led to marked curtailment of Gli1 activation (**Figure 5E,F**). Expression of Smurfs showed a compensatory upregulation in response to the loss of the other Smurf in these MEFs as reported (**Yamashita et al., 2005**; **Tang et al., 2011**), but surprisingly, ShhN induced

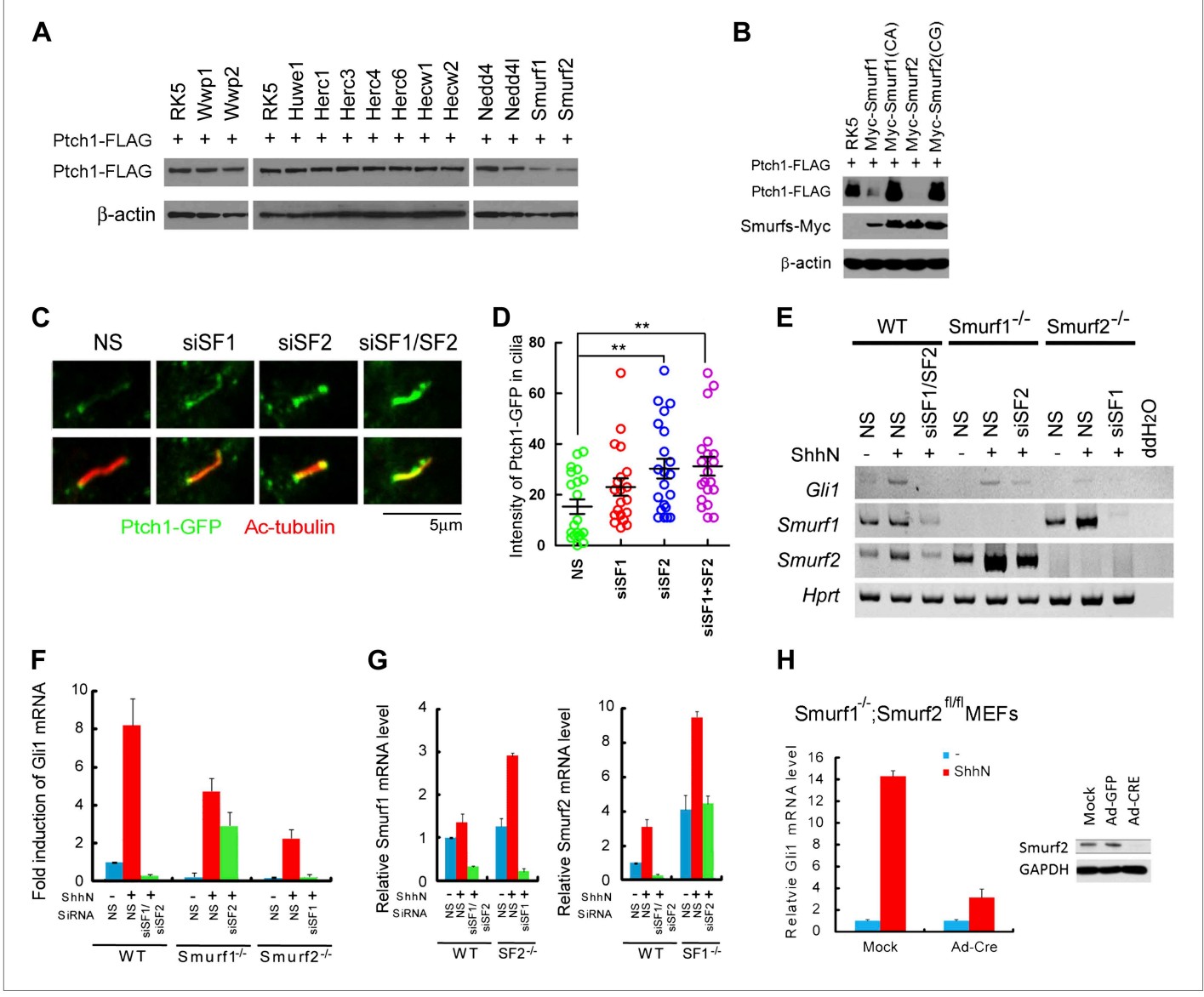

**Figure 5**. Smurf1 and Smurf2 are E3 ligases required for Shh signaling. (**A**) Western analyses of Ptch1-FLAG in HEK293 cells that were co-transfected with cDNAs encoding a battery of HECT-domain E3 ligases as indicated, and (**B**) ligase deficient Smurf mutants. β-actin was used as a loading control. (**C**) Representative confocal images and (**D**) calculations of Ptch1-GFP fluorescence accumulated in primary cilia as the result of siRNA knockdown of *Smurf1*, *Smurf2*, or both in NIH3T3 cells. Primary cilia were marked by acetylated tubulin (red). (**E**) RT-PCR detection of Gli1, Smurf1, and Smurf2 mRNAs in wildtype (WT), *Smurf1⁻/⁻*, and *Smurf2⁻/⁻* MEFs transfected with non-silencing (NS) or Smurf-specific siRNAs as indicated. HPRT mRNA was used as an internal control. A representative gel image is shown here. (**F**) RT-qPCR quantification of fold induction of Gli1 mRNA from an experiment as in (**E**). Fold induction was calculated using Gli1 mRNA level normalized against that of Hprt for even loading and then against the normalized Gli1 mRNA level from cells transfected with NS siRNA and without ShhN treatment. (**G**) RT-qPCR analysis of relative levels of Smurf1 and Smurf2 mRNAs from the experiment in (**F**). (**H**) RT-qPCR detection of endogenous Gli1 mRNAs in *Smurf1⁻/⁻;Smurf2fl/fl* MEFs that were infected with either Ad-GFP (mock) or Ad-Cre for 12 hr, and then treated with either control or ShhN conditional medium for 72 hr. (**I**) Western analyses of endogenous Smurf2 in *Smurf1⁻/⁻;Smurf2fl/fl* MEFs from the experiment in (**H**).

The following figure supplements are available for figure 5:

**Figure supplement 1**. Knockdown of Smurf1 and Smurf2 simultaneously dampens Shh-induced ciliary exit of Ptch1-GFP.

expression of both Smurfs (*Figure 5E,G*). During the course of this investigation, we generated *Smurf1⁻/⁻;Smurf2fl/fl* mice, which will be described in detail elsewhere. In *Smurf1⁻/⁻;Smurf2fl/fl* MEFs, Ad-cre infection-mediated ablation of conditional *Smurf2fl* alleles severely dampened the Gli1

transcriptional response to ShhN (*Figure 5H,I*). Similarly, two other Shh signaling responses, namely Shh-induced ciliary import of Smo and Gli3, were also affected (*Figure 6A–C*). Since we could rescue Shh induction of GliBS-luc responses in Ad-cre infected *Smurf1⁻/⁻;Smurf2ᶠˡ/ᶠˡ* MEFs (*Smurfs* null) by reintroducing wildtype Smurf1 or Smurf2 but not mutant Smurf1CA or Smurf2CG cDNA (*Figure 6D*), or by siRNA-mediated knockdown of *Suppressor of fused* (*Sufu*) (*Figure 6E*), an essential downstream negative regulator of Shh signaling, the observed defects of GliBS-luc induction have to be Smurfs and Shh pathway specific. Taken together, the above results show that simultaneous inactivation of both *Smurf* genes and removal of the 'PPXY' motifs of Ptch1 have congruent effects on various Shh signaling events, and indicate that a common Smurf function is required at a step upstream from the control of the ciliary import of Smo.

## Smurfs and Ptch1 colocalize and interact in lipid rafts

If Smurfs are the E3 ligases that recognize the endocytic sorting signals of Ptch1, these proteins should physically interact in lipid rafts. A number of evidence demonstrates that this is the case. First, in non-permeabilizing MEFs, we found exogenously expressed Ptch1-RFP colocalized with the ligase deficient, GFP-tagged Smurf2CG mutant in Cav-1 positive lipid rafts at the rim of the plasma membrane (*Figure 7A*). Although first identified as modulators of TGF-β/BMP signaling, Smurfs are preferentially localized in the nucleus (*Kavsak et al., 2000*) and play a crucial function in maintaining genomic stability (*Blank et al., 2012*). Serendipitously, we found that treatment with ShhN ligand or co-expression with Ptch1-RFP each caused Smurf2GFP to move from the nucleus to the cytoplasm (*Figure 7B*, *Figure 7—figure supplement 1*). In light of the Shh induction (*Figure 5E,F*), these results indicate that Shh signaling could increase the cytoplasmic pool of Smurfs. Third, fluorescence resonance energy transfer (FRET) analysis showed that Ptch1-CFP was localized in close proximity with Smurf1-YFP or Smurf2-YFP at punctate intracellular vesicles in MEFs (*Figure 7C,D*), and ShhN treatment enhanced this colocalization (*Figure 7E*). However, Δ2PY-CFP failed to generate FRET with Smurf2-YFP (*Figure 7C,D*). Theses result were further corroborated in the discontinuous sucrose gradient sedimentation experiment described earlier, in which the ligase-deficient Smurf2CG-Myc co-sedimented in the Cav-1-containing 20–25% sucrose fractions readily with Ptch1-FLAG, whereas Δ2PY was inefficient in bringing Smurf2CG-Myc into these fractions (*Figure 1E*). Finally, using co-immunoprecipitation, we demonstrated that Ptch1 specifically binds either Smurf1 or Smurf2, and Ptch1 mutants lacking either PY-1 or PY-2 motif can still bind Smurfs, albeit with reduced affinity; however, Δ2PY completely lacks affinity for either Smurf1 or Smurf2 (*Figure 7F*).

## Smurfs are required for Ptch1 turnover and ubiquitin modification

To delineate the requirement of Smurfs for Shh-induced Ptch1 turnover, we took the advantage of the conditional *Smurf1⁻/⁻;Smurf2ᶠˡ/ᶠˡ* MEFs, and quantified the turnover rate of exogenously expressed Ptch1-FLAG following cyclohexamide treatment without or with removal of the *Smurf2* alleles following Ad-cre infection. The results indicated that Ptch1-FLAG was indeed rendered stable against ShhN induced degradation by the removal of the *Smurf2ᶠˡ* conditional alleles whereas the stability of Δ2PY was resistant to change in response to either ShhN treatment or eradication of *Smurf*'s function (*Figure 8A–D*). The induction by Shh is likely a function of ligand-binding, rather than a signaling outcome, as the loop2 mutant Ptch1 that lacks the ability to bind Shh (*Briscoe et al., 2001*) completely lost the capacity to respond to ShhN treatment in wildtype MEFs, although it was more stable in *Smurf*-null MEFs (*Figure 8E,F*). We further found that Shh-induced endocytic turnover of Ptch1 was not affected in *Smo* null MEFs (*Figure 8G,H*), suggesting that it is an upstream signaling event, independent of Smo function.

To demonstrate the Ubiquitin E3 ligase activity of Smurfs on Ptch1, we assayed for the ability of Ptch1-FLAG or Δ2PY to be ubiquitinated by HA-tagged Ubiquitin (HA-Ub) in *Smurf1⁻/⁻;Smurf2ᶠˡ/ᶠˡ* MEFs. In these cells, Ptch1-FLAG was readily ubiquitinated, but the level of ubiquitination of Ptch1Δ2PY-FLAG was diminished (*Figure 9A*). More importantly, neither of the two forms of Ptch1 was ubiquitinated after the conditional *Smurf2ᶠˡ* alleles were removed with Ad-cre, (*Figure 9A*). We were also able to demonstrate ubiquitination of Ptch1-FLAG that was produced and isolated from HEK293 cells in an in vitro reconstituted system, in which the level of ubiquitinated species was greatly enhanced by His6-Smurf2, but not the ligase-inactive His6-Smurf2CG purified from the insect expression system (*Figure 9B*), indicating a direct enzyme and substrate relationship. Although we were not able to detect mono-ubiquitination, the poly-ubiquitin chains on Ptch1 are likely of both K48 and K63 linkage,

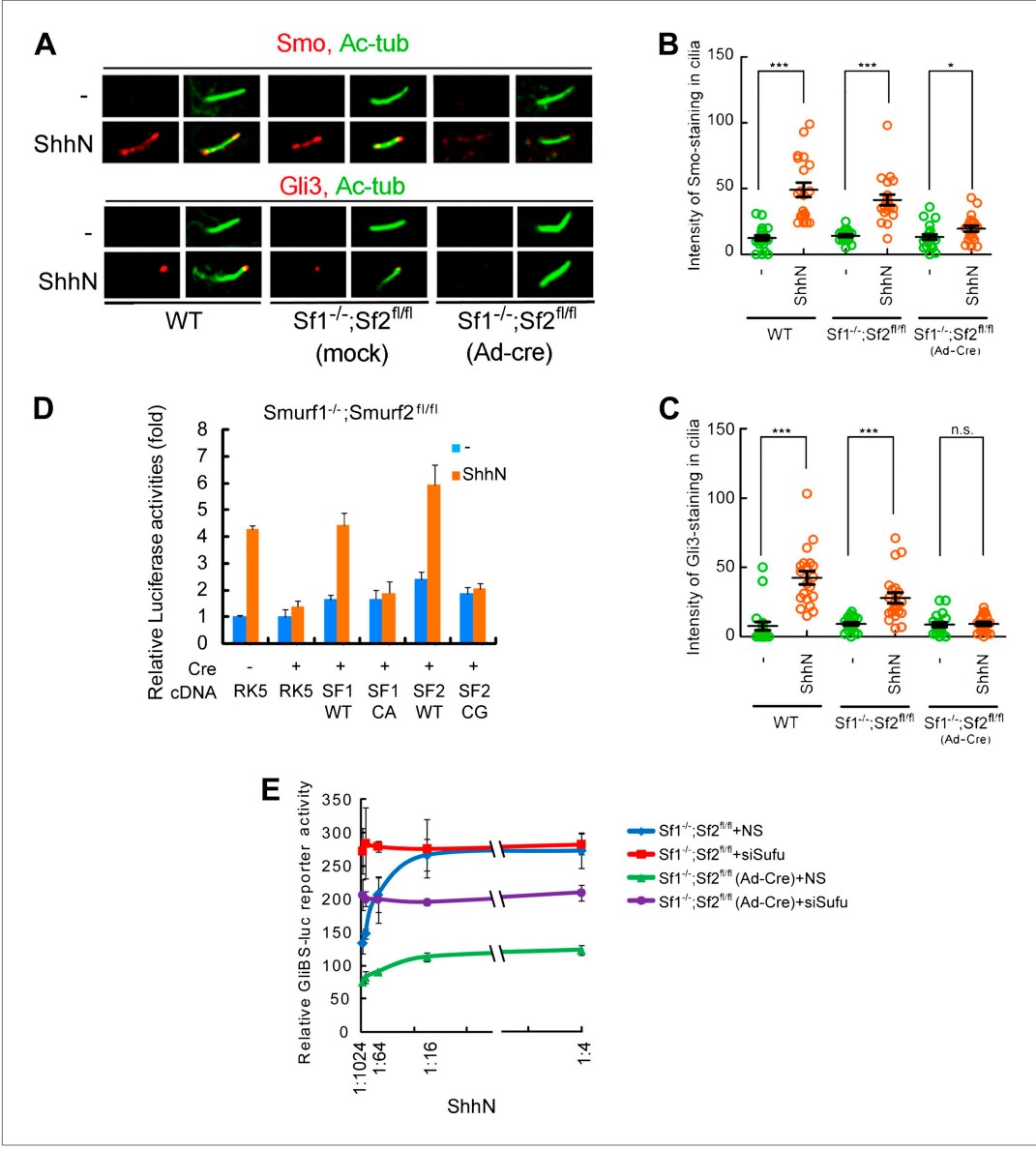

**Figure 6**. Smurf1 and Smurf2 are required For Shh signaling. (**A**) Representative confocal images of Smo and Gli3 immunofluorescence staining in cilia of wildtype (WT), *Smurf1⁻/⁻;Smurf2ᶠˡ/ᶠˡ*, or *Smurf1⁻/⁻;Smurf2ᶠˡ/ᶠˡ* MEFs infected with Ad-Cre viruses. (**B**) Quantification of Smo and (**C**) Gli3 immunofluorescence staining in cilia of (**A**). In the above experiments, ShhN treatment was carried out for 24 hr, and the means and standard deviation were calculated from two independent experiments (n = 20). (**D**) GliBS-luc assays in *Smurf1⁻/⁻;Smurf2ᶠˡ/ᶠˡ* MEFs showing the deficiency of Shh induction associated with genomic ablation of both Smurfs can be rescued by re-introducing either wildtype Smurf1 or Smurf2 but not their corresponding mutants. (**E**) GliBS-luc reporter assays for the ability of siSufu to by-pass the requirement of Smurfs in Shh signaling. *Smurf1⁻/⁻;Smurf2ᶠˡ/ᶠˡ* MEFs were infected with Ad-cre and then transfected with siSufu or ns control. The cells were then treated with a series of dilutions of ShhN-CM before luciferase activities were assayed. Error bars denote standard deviations. Statistical analyses were performed by two-tail Student's *t* test. *p<0.05, **p<0.01, ***p<0.001, and *n.s.*, not significant (p>0.05).

as re-expression of Smurf2-Myc in *Smurf2* null cells enhanced Ptch1 ubiquitination in the presence of wt, KO, K48, or K63 ubiquitin (**Figure 9C**). Finally, ShhN treatment enhanced the level of high molecular weight ubiquitinated Ptch1 species in wildtype MEFs (**Figure 9D**), consistent with the ability of Shh to induce Ptch1 turnover (**Figure 8**).

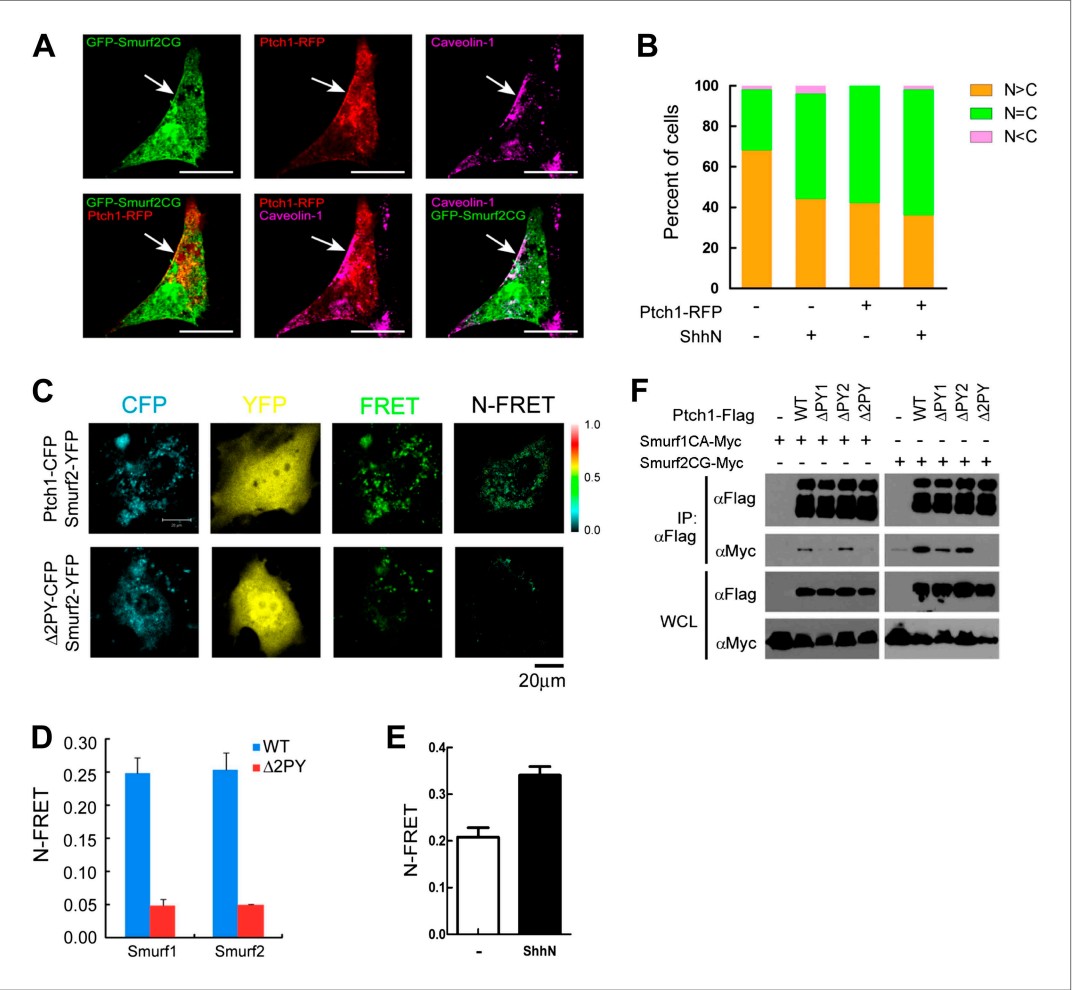

**Figure 7**. Colocalization and interaction between Ptch1 and Smurfs in Cav-1 positive lipid rafts. (**A**) Confocal images showing colocalization of GFP-Smurf2CG and Ptch1-RFP in Cav-1 positive lipid rafts. The cells were not permeabilized before they were stained with anti-Cav-1, and the images were taken with a 63x oil lens. (**B**) Quantification of nuclear and cytoplasmic distribution of Smurf2GFP as in *Figure 7—figure supplement 1*. The percentage of mostly nuclear (N > C), even distribution (N = C), or mostly cytoplasmic (N < C) of Smurf2GFP pattern cells was calculated based images of 40 cells at each data point. (**C**) FRET analysis of Ptch1-CFP or Δ2PY-CFP interaction with Smurf1-YFP or Smurf2-YFP in transfected MEFs. Representative images of CFP, YFP, FRET fluorescence, and N-FRET are shown. (**D**) Quantification of N-FRET values using the sensitized emission method, which is expressed as means plus SD in the bar graph. (**E**) FRET analysis of Ptch1-CFP interaction with Smurf2-YFP in transfected MEFs that were treated with ShhN or control conditioned medium for 2 hr. Quantification of N-FRET values described in (**D**). (**F**) Co-immunoprecipitation analyses of FLAG-Ptch1 and the 'PPXY' motif mutants with Myc-tagged Smurf1CA or Smurf2CG ligase-deficient mutants. The immunocomplexes were precipitated using anti-FLAG, and blotted with anti-Myc.

The following figure supplements are available for figure 7:

**Figure supplement 1**. ShhN treatment and co-expression with Ptch1 caused Smurf2 to redistribute from the nucleus to the cytoplasm.

**Figure supplement 2**. Neither Smurf1 nor Smurf2 interact with Smo.

## Requirement of Smurfs in sustaining the proliferation of cerebellar granule cell precursors by Shh

Mice deficient in both *Smurf1* and *Smurf2* were reported embryonic lethal due to absence of planar cell polarity among other pleiotropic defects (*Narimatsu et al., 2009*). More than half of the double

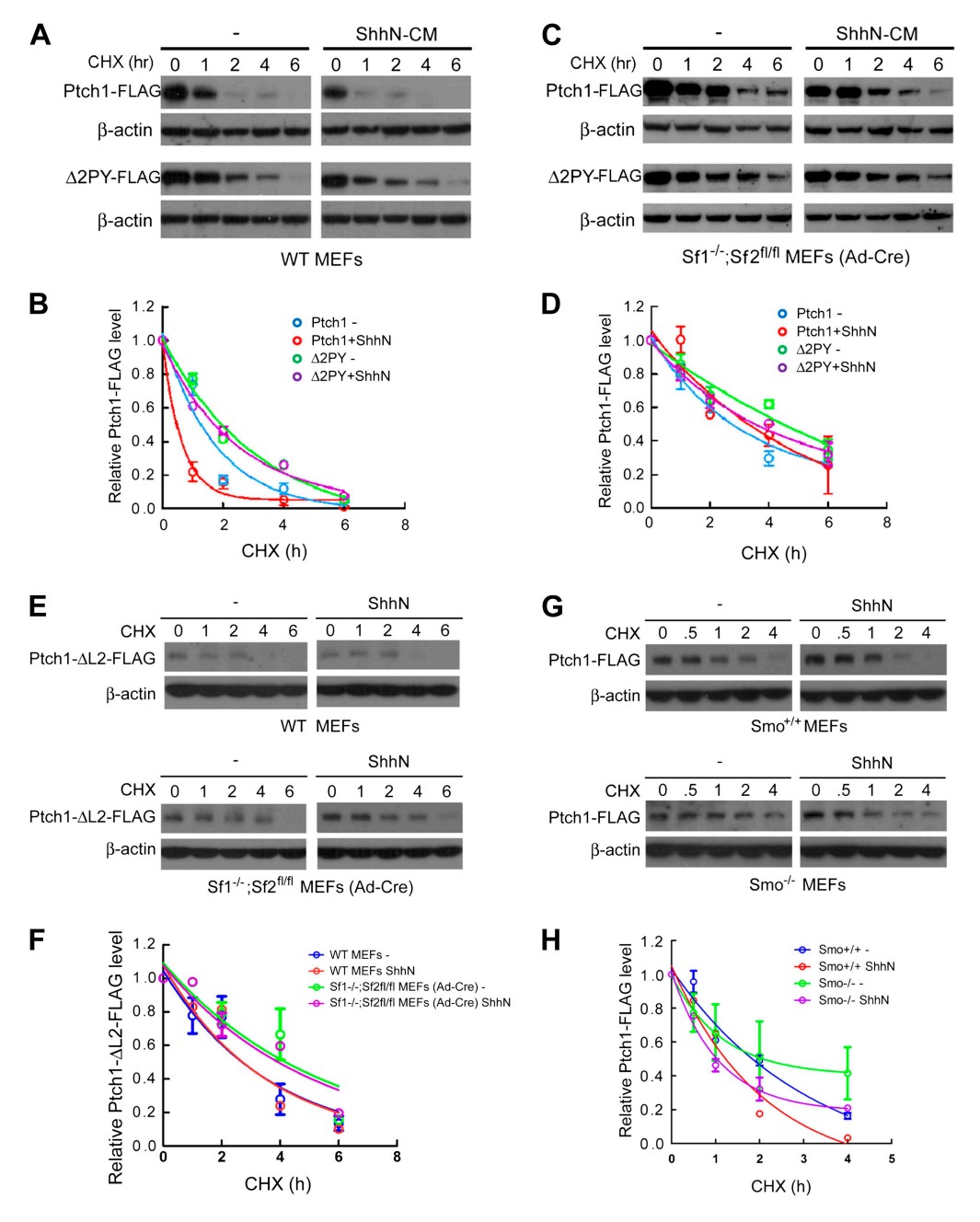

**Figure 8**. Smurfs are required for the Shh-induced endocytic turnover of Ptch1. Western analysis of Ptch1-FLAG and Δ2PY-FLAG turnover rates (**A**) and quantification thereof (**B**) in WT MEFs. ShhN and CHX were added for duration as indicated. (**C**) Western analysis of Ptch1-FLAG and Δ2PY-FLAG turnover rates (**C**) and quantification thereof (**D**) in *Smurf1⁻/⁻;Smurf2ᶠˡ/ᶠˡ* MEFs infected with Ad-cre. (**E**) Western analysis of Ptch1-Δloop2-FLAG turnover rate and quantification thereof (**F**) in WT (upper) and *Smurf1⁻/⁻;Smurf2ᶠˡ/ᶠˡ* MEFs infected with Ad-cre (lower). (**G**) Western analysis of Ptch1-FLAG turnover rate and quantification thereof (**H**) in WT (upper) and *Smo⁻/⁻* MEFs (lower). Each data point denotes mean ± standard deviation from two independent experiments.

null embryos that we generated failed to gastrulate and those rare embryos that did escape seldom passed Theiler stage 13, thus precluding a thorough analysis of the neural tube phenotype where Shh function is well characterized. To address the physiological relevance of Smurf regulation of Ptch1 endocytosis, we examined the role of Smurfs in sustaining the proliferation of cerebellar granule

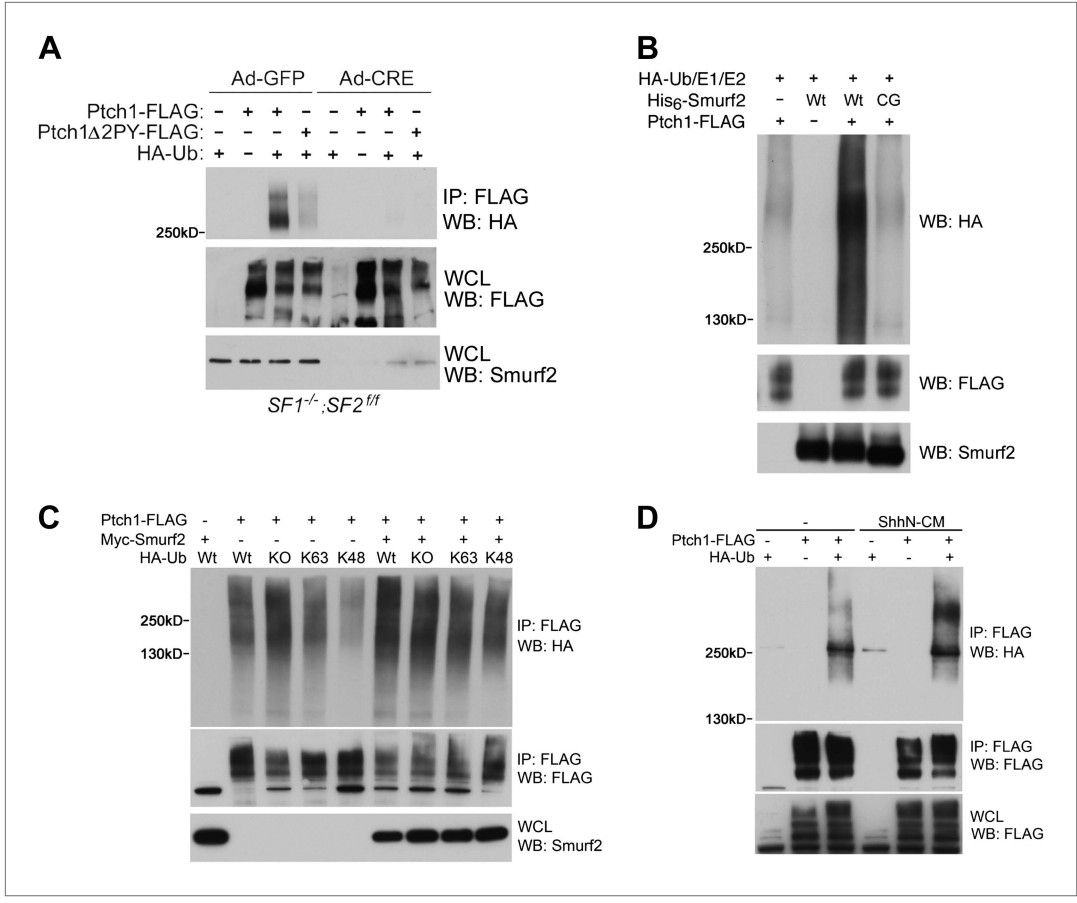

**Figure 9**. Smurfs are required for ubiquitin modification of Ptch1. (**A**) Western analysis of ubiquitinated Ptch1-FLAG and Ptch1Δ2PY-FLAG in *Smurf1⁻/⁻;Smurf2ᶠˡ/ᶠˡ* MEFs infected with Ad-GFP or Ad-Cre. These MEFs were first infected with adenoviruses and then transfected with HA-Ub and the Ptch1 plasmids as marked. The exogenously expressed Ptch1 proteins were immunoisolated using anti-FLAG beads prior to analysis. (**B**) Western analysis of Ptch1-FLAG ubiquitination in vitro in a reconstituted system comprising purified recombinant His₆-Smurf2 or the ligase-deficient His₆-Smurf2CG from the baculovirus, HA-Ub, and an ATP regeneration system. Ptch1-FLAG was immunoisolated from HEK293 cells and the ubiquitination reaction was carried out on beads. The proteins were eluted prior to Western blot analysis. (**C**) Western analysis of ubiquitinated Ptch1-FLAG in Smurf2⁻/⁻ MEFs that were also transfected with Wt, KO, K48, or K63 ubiquitin in the absence or presence of Myc-Smurf2. (**D**) Western analysis of ubiquitinated Ptch1-FLAG in WT MEFs treated with ShhN or control conditioned medium. Ptch1-FLAG in A-C was resolved by 6% SDS-PAGE, but a 4–12% gradient gel was used in **D**.

cell precursors (GCPs), which has an absolute requirement for Shh. For this purpose, we cut cerebellar slices from P7 *Smurf1⁻/⁻;Smurf2ᶠˡ/ᶠˡ* pups and cultured them for 12 days in vitro as described (*Kapfhammer, 2010*). Anti-NeuN immunofluorescence staining revealed that the number of postmitotic granule cells were severely reduced in slices infected with Ad-cre viruses (*Figure 10A*), suggesting that Shh signaling was compromised there. We also isolated GCPs from cerebella of normal P7 pups of the C57/B6 strain, and cultured them in vitro. In the presence of ShhN, GCPs grew healthily for at least 5 days, but siRNA knockdown of *Smurf1* and *Smurf2* simultaneously blocked GCP proliferation (*Figures 10B*, *Figure 10—figure supplement 1A,C*). To ascertain that the effect of Smurf knockdown was Shh-pathway specific, we repeated the above experiment using IGF1, which is capable of sustaining the proliferation of GCPs in lieu of Shh (*Rao et al., 2004*; *Fernandez et al., 2010*), and found that knockdown of *Smurfs* had no effect on IGF-1-induced GCP growth (*Figure 10C*, *Figure 10—figure supplement 1B*). Thus, these data unequivocally demonstrated that Smurf1 and Smurf2 share a critical role in supporting Shh signaling during cerebellar organogenesis.

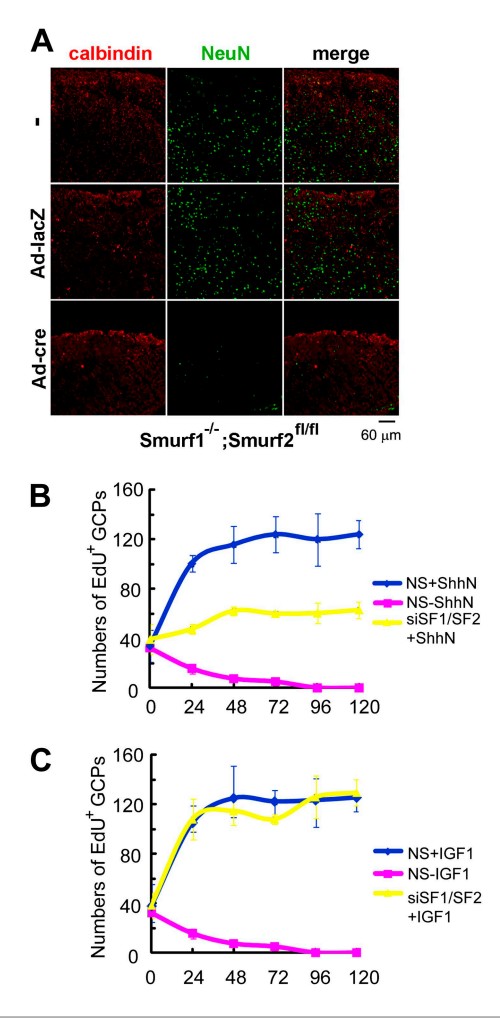

**Figure 10**. Requirements of Smurfs for Shh-induced organogenesis. (**A**) Immunostaining of P7 cerebellar slices cultured in vitro with anti-calbindin (red) and anti-NeuN (green). The slices were first infected with control or cre-expressing adenoviruses for 24 hr and then continuously cultured for 12 days. Quantification of EdU incorporated GCPs in cerebellar slices cultured in the presence of ShhN from *Figure 10—figure supplement 1A* (**B**) or IGF-1 from *Figure 10—figure supplement 1B* (**C**), respectively. The data at each time point were derived from four separate fields, and the bars denote standard deviation.

The following figure supplements are available for figure 10:

**Figure supplement 1**. Smurfs are required for ShhN but not IGF-1 induced GCP proliferation.

## Discussion

Shh plays a fundamental role in setting up the body plan during embryogenesis, and is also critical in guiding stem cell differentiation for maintaining tissue homeostasis in the adult. Cell surface reception of Shh signaling is a multistep process that entails, but is not limited to, ligand engagement, reciprocal movements of Ptch1 exiting from and Smo translocating into the primary cilium, and activation of the G-protein-coupled Smo by still-controversial mechanisms (*Ogden et al., 2008*). The central task of this process is to sense and convert incremental changes in the Shh gradient into corresponding levels of signaling output, thereby allowing the positional cues to be executed. In this study, we extended our knowledge of the Shh signaling activation process by revealing a ubiquitination switch that regulates Ptch1 endocytosis, which is essential in clearing Ptch1 from its site of action in the primary cilium, and to ligand sequestration, as previously described (*Incardona et al., 2000*). Our data demonstrate that ubiquitination of Ptch1 mediated by the two 'PPXY' motifs is controlled by HECT-domain E3 ligases Smurf1 and Smurf2, which are induced by Shh (*Figure 5E,G*) and redistributed into the cytoplasm under Shh influence (*Figure 7B*, *Figure 7—figure supplement 1*). Shh also promotes the association of Ptch1 and Smurfs in intracellular vesicles (*Figure 7E*), most likely the Cav-1 positive lipid rafts (*Figure 1A,B*), as well as ubiquitination (*Figure 9D*) and endosomal entry (*Figure 2A–C*), leading to lysosomal turnover (*Figures 1F, 8A–D*). So, an increase in the Shh signal strength would cause a corresponding increase in both the production of Ptch1 and its rate of turnover en route from the primary cilium to the lipid rafts and to the endosomes/lysosomes. This regulatory scheme is reminiscent of an electronic amplification circuit, in which a feedback loop added to an open-loop amplifier has the effect of stabilizing the gain and increasing the linearity of the output signal to a given range, which can be controlled by adjusting the feedback strength. By analogy, Shh induction of Gli1 can be viewed as the open-loop amplifier, with Ptch1 providing the negative feedback. In this wiring logic, the graded Shh morphogenic signal can be stably transformed into stepwise output responses tailored for a predetermined cell fate specification. Without endocytosis, Ptch1 would

accumulate in the primary cilium (*Figure 1A,B*, *Figure 1—figure supplement 2*, *Figure 3—figure supplement 1*), thus hampering Smo import and function. More importantly, without Ptch1 removal/degradation, the amplitude of Shh signaling would be restricted by the accumulation of newly synthesized inhibitory Ptch1. Oversupplied Ptch1 could also impact on signaling in neighboring cells

through non-cell autonomous inhibition. So, Ptch1 endocytosis plays a crucial role in setting the output range of Shh signaling.

The presence of Ptc in membranous vesicles has long been noted in *Drosophila* and mammalian cells (*Capdevila et al., 1994*; *Denef et al., 2000*; *Ramirez-Weber et al., 2000*; *Zhu et al., 2003*), but its significance was not fully appreciated and regulation unknown. Ptc or Ptch1 is a 12-pass transmembrane protein, whose internal sequence spanning from IV to X transmembrane domains resembles the resistance, nodulation, division (RND) family of bacterial proton-driven transporter and the sterol-sensing domain found in SREBP and NPC1 (*Carstea et al., 1997*; *Taipale et al., 2002*). Substantial evidence in the literature suggests that Ptch1 inhibition of Smo occurs by way of small molecular intermediates that may be transported by Ptch1 through the membrane (*Di Guglielmo et al., 2003*; *Bijlsma et al., 2006*; *Yavari et al., 2010*). Perhaps it is not a coincidence that we found Ptch1 exits the primary cilium and enters the endocytic pathway via cholesterol and sphingomyelin-rich lipid rafts, whereas Smo was shown previously to enter the primary cilium via Clathrin-coated pits when induced by Shh (*Chen et al., 2004*; *Kovacs et al., 2008*). It is possible that Ptch1 and Smo are required to be sorted into different membranous compartments and to keep a mutually exclusive presence in the primary cilium, so that a cross-membrane imbalance of the small molecular intermediates is attained. The RND/sterol-sensing domain is critical to Ptch1 function as multiple inactivating mutations in this region have been found in *Drosophila* as well as in Gorlin syndrome patients (*Martin et al., 2001*; *Strutt et al., 2001*; *Taipale et al., 2002*). However, although certain RND mutants of *Drosophila* Ptc accumulate in endosomes (*Martin et al., 2001*; *Strutt et al., 2001*), this domain may be more important to Ptch1 function than to its endocytic turnover, since we found that combining a RND mutation with the 2-PY deletion did not alter the latter's impact on Ptch1 stability (data not shown).

Through cDNA-mediated screens, we have identified Smurf1 and Smurf2 as the E3 ligases responsible for generating the sorting signal for Ptch1 endocytosis. Although subsequent experiments indicated that deletion of one *Smurf* gene was not sufficient to inactivate Shh signaling, siRNA-mediated knockdown of either *Smurf1* or *Smurf2* was enough to dampen the 8xGliBS reporter response in transfected MEFs. This apparent discrepancy is likely to be reconciled by the mutual, compensatory upregulation of either of the two *Smurf* genes upon the loss of the other, resulting in the adaptation of single-*Smurf*-knockout MEFs for a robust Shh signaling response. On the other hand, such an adaptive response might not have been established in time under the conditions found in transiently transfected MEFs in response to siRNA-mediated knockdown. The observation of Shh induction of Smurf expression (*Figure 5E,G*) and cytoplasmic pivoting (*Figure 7B*, *Figure 7—figure supplement 1*) further implicated Smurfs in Shh signaling. Previously, *Drosophila* Ptc was shown to interact with and regulated by Nedd4 (*Lu et al., 2006*), another HECT-domain E3 ligase. In addition, the mouse Ptch1 was also shown to bind Nedd4, but this interaction triggers apoptosis through ubiquitination of Caspase 9 (*Fombonne et al., 2012*). It is likely that Ptch1 is regulated by multiple E3 ligases with different functional outcomes. Recently, *Drosophila* DSmurf was identified as a Ptc-interacting partner in a yeast 2-hybrid screen, and shown subsequently as a specific E3 ligase that regulates Ptc stability (*Huang et al., 2013*). However, DSmurf was shown to promote Ptc turnover in the presence of activated Smo$^{SD}$, bind Smo, and prefer ligand-unbound Ptc as a substrate (*Huang et al., 2013*). We did not observe interaction between mammalian Smurfs and Smo by Co-IP experiments (*Figure 7—figure supplement 2*), and found that Shh induction of Ptch1 turnover proceeded unabatedly even in the absence of Smo (*Figure 8G,H*). In Huang et al., when ectopically expressed in the anterior compartment of the wing disc, activated Smo$^{SD}$ induced massive amount of Ptc; these two proteins could form a complex at the high levels, much like their mammalian counterparts do when overexpressed in HEK293 cells (*Stone et al., 1996*; *Taipale et al., 2002*). Perhaps, DSmurf could recognize this unnatural complex and triggers a proteasomes-mediated degradation, even specifically.

Smurf2 was shown previously to function in lipid rafts (*Di Guglielmo et al., 2003*), and the necessity of removing both *Smurf1* and *Smurf2* to reveal their requirement in Shh signaling strongly argues that this shared function has a deep root in evolution. In any event, our work presents a rather comprehensive view of the Shh pathway activation process. Considering two neighboring cells in a given Shh influence field (*Figure 11*), the cell that receives lower Shh input (upper cell) encounters a stronger feedback inhibition due to lower endocytic turnover of Ptch1, resulting in a lower level of Shh signaling output represented by Gli1. In the cell that receives higher Shh input (lower cell), although the synthesis of Ptch1 is induced, upregulation of Smurfs and the induction of colocalization in lipid rafts ensure a faster Ptch1 turnover such that the level of Ptch1 feedback inhibition is actually low, resulting in higher

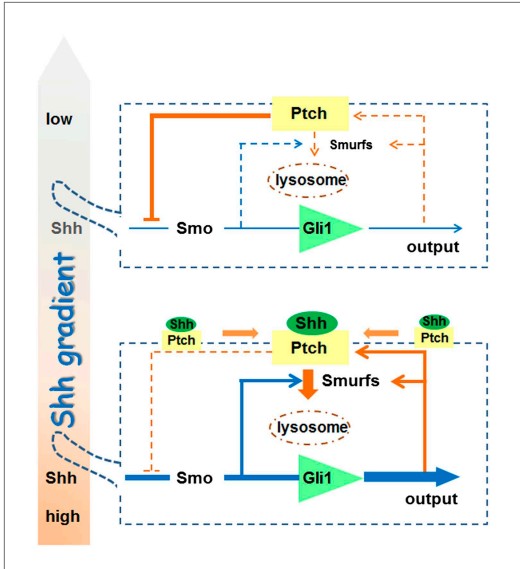

**Figure 11**. A model for the role of Smurf-mediated Ptch endocytosis in Shh signaling.

pathway activity. The endocytic turnover also has impact on the ligand sequestration role of Ptch1 through controlling the availability of the ligand 'sink' on cell surface. In this regard, the Smurf-mediated endocytosis of Ptch1 is an essential signaling event, and it is theoretically possible to block Shh function both cell and non-cell autonomously using Smurf inhibitors, thus opening a new route for Shh-targeted cancer treatment.

# Materials and methods

## Animals

All mice were maintained and handled according to protocols approved by the Animal Care and Use Committee of the National Cancer Institute, NIH. The conditional *Smurf2* knockout allele, *Smurf2^fl* was generated by insertion of two loxP sites into introns flanking Exon 9 and 10 through homologous recombination. Further details of the construction will be described elsewhere.

## Cells, plasmids, and siRNAs

*Smurf1^−/−*, *Smurf2^−/−*, and *Smurf1^−/−;Smurf2^fl/fl* MEFs were isolated from E14.5 embryos and cell immortalization was carried out according to the 3T3 protocol. NIH3T3:Gli-Luc-3T3 and *Ptch1^−/−* MEFs were described previously (*Chen et al., 2011*). Full-length mouse Ptch1 cDNA was obtained from ATCC, and the FLAG, GFP, or RFP-tagged variants of which were generated by PCR and subcloned into the pRK5 vector. The ΔPY mutants of Ptch1 were generated using a PCR-based strategy. All PCR-amplified fragments were sequence verified. Plasmids for Myc-tagged Smurf1, Smurf1CA, Smurf2, Smurf2CG, GFP-tagged Smurf2, HA-tagged Ub, UbKO, UbK63 and UbK48 were described previously (*Zhang et al., 2001*; *Yamashita et al., 2005*, *2008*; *Tang et al., 2011*; *Blank et al., 2012*). RFP-tagged Rab5, Rab7, and Lamp1 were acquired from Addgene. siRNAs specific for the mouse HECT family of E3 ligases and cDNAs encoding human HECT E3 ligases were purchased from QIAGEN (Germantown, MD).

## Immunofluorescence staining

Approximately $0.6 \times 10^5$ cells per well were seeded in Lab-Tek chambered slides and cultured for 24 hr. The cells were transfected, allowed to recover for 24 to 36 hr, and then treated with ShhN-CM or other compounds, as indicated. For visualizing ciliary proteins, the transfected cells were starved in DMEM containing 0.5% FBS for 24 hr before other treatments. The cells were fixed with 4% paraformaldehyde for 10 min at 4°C, and standard procedures for immunostaining were followed. The primary antibodies used were rabbit anti-Caveolin-1 (1:1000; Sigma-Aldrich (St. Louis, MO)), rabbit anti-Clathrin heavy chain (1:200; Cell Signaling Technology (Danvers, MA)), rabbit anti-Rab5 (1:150, Cell Signaling Technology), rabbit anti-Rab7 (1:50, Cell Signaling Technology), rabbit anti-Lamp1 (1:150; Sigma), mouse anti-acetylated Tubulin (1:2000; Sigma), rabbit anti-Gli3 (1:500; R&D (Minneapolis, MN)), and rabbit anti-Smo (1:500; a gift from Dr Rajat Rohatgi). Alexa-coupled secondary antibodies were purchased from Life Technologies Corp.

## Confocal microscopy and FRET

Confocal images were acquired on a Carl Zeiss LSM710 microscope. Colocalization coefficient was calculated using Zeiss ZEN2011 program, and quantification of the fluorescence intensity of Ptch1-GFP, Smo, and Gli3 in primary cilia was carried out using Image-Pro as described previously (*Chen et al., 2011*). For FRET analysis, MEFs were transfected with the plasmids encoding Ptch1-CFP or Δ2PY-CFP together with Smurf1-YFP or Smurf2-YFP. Confocal images were acquired with a 40 × objective lens. In track I, cells were excited with a 405-nm laser, and CFP signals were collected in channel II at 470–500 nm. FRET signals were collected in channel III at >530 nm. In track II, cells were excited with a 514-nm laser line, and YFP signals were collected in channel III at >530 nm. FRET efficiency

between CFP and YFP, shown as N-FRET, was calculated using Zeiss ZEN2011 program, and the sensitized emission crosstalk coefficients were determined using control cells that expressed only CFP or YFP.

## GliBS-luc reporter assay for non-cell autonomous inhibition of Ptch1

*Ptch1$^{-/-}$* MEFs were transfected with Ptch1-GFP or Ptch1Δ2PY-GFP along with the Rellina control (15:1) using Lipofectamine Plus (Life technologies, Grand Island, NY)). These cells were then re-seeded with NIH3T3:GliBS-luc reporter cells at 5:1 ratio. After 24 hr, the cells were treated with ShhN-CM in different dilutions for additional 36 hr before the luciferase activities were assayed using the luciferase assay system on a GloMax-96 luminometer (Promega, Madison, WI). The firefly luciferase activity from the indicator cells was normalized against the Rellina luciferase activity to correct for transfection efficiency of Ptch1 constructs in the testing *Ptch$^{-/-}$* MEFs as the measurement of non-cell autonomous inhibition by Ptch1.

## Immunoprecipitation and immunoblotting

Transfected cells were lysed in modified RIPA buffer (50 mM Tris–HCl, pH 7.4, 150 mM NaCl, 1% vol/vol NP-40, 1% n-Dodecyl β-D-maltoside, 0.25% wt/vol sodium deoxycholate, 1 mM DTT, and 1 × Roche cOmplete Protease Inhibitor Cocktail) for 1 hr at 4°C. The lysate was clarified by centrifugation for 1 hr at 20,000×*g*. The protein concentration was determined using a bicinchoninic acid assay and equal amounts of total protein from each of the samples was supplemented with 6 × SDS loading buffer, incubated at room temperature for 1 hr, subjected to SDS-PAGE, followed by western blot analysis. To assay for interactions between exogenous Ptch1-FLAG and the Myc-Smurfs, transfected Ptch1-FLAG was immunopurified with anti-FLAG M2 agarose beads (Sigma) and subjected to SDS-PAGE, followed by western blotting with anti-Myc (Santa Cruz Biotechnology, Dallas, TX).

## Ubiquitination assays

To assay for Ptch1 ubiquitination in vivo, *Smurf1$^{-/-}$/Smurf2$^{flox/flox}$* MEFs were infected with either Ad-GFP or Ad-Cre adenovirus for 24 hr, then transfected with Ptch1-FLAG or Ptch1Δ2PY-FLAG along with HA-Ub using Lipofectamine Plus (Invitrogen). The cells were lysed 24 hr later and Ptch1 and its mutant were isolated with anti-FLAG agarose beads and resolved by SDS-PAGE on 6% or 4–12% gradient gels. The ubiquitinated Ptch1 was then detected with anti-HA (Roche-Shanghai, China). To assay for Ptch ubiquitination in vitro, an ubiquitination assay was modified from a previously described procedure (*Tang et al., 2011*). Ptch1-FLAG was captured from transfected HEK293 cell lysates using anti-FLAG agarose. After a thorough wash, the Ptch1-bound agarose was divided into three aliquots. Empty anti-FLAG agarose was used as a control. The in vitro ubiquitination assay was performed by incubating either Ptch1-bound agarose or control agarose at 37°C for 1 hr with ubiquitin-activating enzyme UBE1, E2-conjugating enzyme UbcH5c, HA-Ub and ATP (all from Boston Biochem, Cambridge, MA) in the presence or absence of purified His6-Smurf2 or His6-Smurf2CG. After the incubation, the supernatant was removed, the agarose thoroughly washed, and the Ptch1-FLAG eluted using the FLAG peptide (Sigma). The eluted fraction was then subjected to Western blot analysis.

## Sucrose gradient sedimentation

Sucrose equilibrium density gradient sedimentation experiments were performed as described (*Coulombe et al., 2004*). Briefly, HEK293 cells grown in 10 cm plates were transiently transfected with Ptch1-FLAG or Δ2PY-FLAG along with Myc-Smurf2CG. 48 hr after transfection, the cells were lysed in pre-chilled 2 ml MES buffer, which contains 25 mM MES (2-[N-morpholino]ethanesulfonic acid), pH 6.5, 150 mM NaCl, 1% Triton X-100, supplemented with 1 × Roche cOmplete Protease Inhibitor Cocktail and was set on ice for 1 hr. The lysates were mixed with equal volume of 80% (wt/vol) sucrose/MES solution and placed at the bottom of an ultracentrifuge tube. Tube was then overlaid in consecutive order with 2 ml each of 30%, 25%, 20%, and 4 ml of a 5% (wt/vol) sucrose/MES buffer. After centrifugation at 39,000 rpm for 16 hr at 4°C in an SW 41 Ti rotor on Beckman Optima L-100 XP ultracentrifuge, the gradient was separated into twelve 1 ml fractions taken from the top for Western blot analysis.

## RT-PCR and quantitative real-time PCR

Total RNA was isolated from cultured cells using the RNAiso reagent (TaKaRa, Shiga, Japan), and reverse transcription was carried out using the PrimeScript RT reagent Kit (TaKaRa). Standard RT-PCR was carried out with the following primers: mouse Gli1 (5′-TCCAGCTTGGATGAAGGACCTTGT-3′ and 5′-AGCATATCTGGCACGGAGCATGTA-3′), mouse Smurf1 (5′-CTACCAGCGTTTGGATCTAT-3′

and 5'-TTCATGATGTGGTGAAGCCG-3'), mouse Smurf2 (5'-TAAGTCTTCAGTCCAGAGACC-3' and 5'-AATCTCTTCCCTAGACACCTC-3'), and mouse HPRT (5'-TATGGACAGGACTGAAAGAC-3' and 5'-TAATCCAGCAGGTCAGCAAA-3'). Real-time PCR was carried out using the FastStart SYBR Green Master mix (Roche) on a 7500 Real-Time PCR System (Applied Biosystems, Grand island, NY) with primers for mouse Gli1 (5'-GCTTGGATGAAGGACCTTGTG-3' and 5'-GCTGATCCAGCCTAAGGTTCTC-3') and mouse HPRT (5'-TATGGACAGGACTGAAAGAC-3' and 5'-TAATCCAGCAGGTCAGCAAA-3'). Experiments were repeated at least three times, and samples were analyzed in triplicate.

## Cerebellar slice culture

Cerebellar slice cultures were prepared as described (*Kapfhammer, 2010*). Briefly, sagittal sections (350 µm) were cut from cerebella of P7 *Smurf1$^{-/-}$;Smurf2$^{fl/fl}$* pups using a McIlwain tissue cutter under septic condition. Slices were transferred onto a permeable membrane (Millicell-CM, Millipore-China, Beijing, China) in a 6-well plate with 0.8 ml of culture medium (Neurobasal A medium with B27 supplement) and incubated at 37°C, 5% CO2. For adenovirus infection, the viral stock (3 × 10$^{10}$ pfu/ml) was mixed with equal volume of type I collagen gel and applied as a drop on top of each slice, and 5 × 10$^7$ pfu of virus was also added in the culture medium. After 24 hr, the infected slices were washed and maintained in culture medium. The medium was changed every 2–3 days for a total of 12 days. Slices were then fixed in 4% paraformaldehyde overnight at 4°C and immunostained with anti-calbindin (1:500; Sigma) and anti-NeuN (1:100; Millipore).

## GCP isolation and proliferation assay

Mouse cerebellar GCPs were isolated from 7-day-old pups according to a published protocol (*Hatten and Shelanski, 1988*). Briefly, cerebella were removed aseptically and incubated at 37°C for 5 min in trypsin/DNase buffer. Tissues were then triturated with fine Pasteur pipettes to obtain a single-cell suspension, overlaid on top of a step gradient of 35% and 65% Percoll (Pharmacia, GE Health-China, Shanghai, China) and centrifuged at 2,000×g for 10 min at 4°C. GCPs harvested from the 35% and 65% Percoll interface were further purified by depleting adherent cells with two consecutive 1-hr incubations in tissue culture dishes, then seeding them in Lab-Tek chambered slides coated with poly-D-lysine and Matrigel, and incubating them at 35°C, 5% CO$_2$. GCPs were transfected with siRNAs using FugeneHD Transfection Reagent (Promega) after 1 hr incubation. Proliferation of transfected GCPs was evaluated using Click-iT EdU cell proliferation assays (Life Technologies) at different time points after ShhN-CM or IGF1 (100 ng/ml) treatment. GCPs were incubated with EdU (5-ethynyl-2'-deoxyuridine) for 12 hr before fixation and permeabilization. EdU detection was performed according to the manufacturer's instruction. Images were acquired on a Leica inverted fluorescence microscope (DMI 300B) with a 20 × objective lens. Quantification of EdU-positive GCPs was performed using the ImageJ software.

## Acknowledgements

We wish to thank Rajat Rohatgi for the generous gift of the Smoothened antibody, and Tian Jin, Joseph Brzostowski and Valarie Barr for their assistance with confocal imaging. This work was supported by funding from the US-China Biomedical Collaborative Research program to SYC and YEZ; grants from the Chinese National Science foundation (81272238 and 81261120386) and the National Basic Research Program of China (973 Program) to SYC (2012CB945003 and 2009CB918403); and by funding from the intramural research program of the National Institutes of Health, National Cancer Institute, Center for Cancer Research to YEZ. SY is supported by a young investigator grant from the Chinese National Science Foundation (81101497).

## Additional information

### Funding

| Funder | Grant reference number | Author |
| --- | --- | --- |
| National Natural Science Foundation of China (NSFC) | 81272238, 81261120386 | Steven Y Cheng |
| Ministry of Science and Technology of the People's Republic of China (Chinese Ministry of Science and Technology) | 2012CB945003, 2009CB918403 | Steven Y Cheng |

| Funder | Grant reference number | Author |
|---|---|---|
| National Institutes of Health (NIH) | ZIA BC 011168 | Ying E Zhang |
| National Natural Science Foundation of China (NSFC) | 81101497 | Shen Yue |

The funders had no role in study design, data collection and interpretation, or the decision to submit the work for publication.

## Author contributions

SY, Acquisition of data, Analysis and interpretation of data, Drafting or revising the article; L-YT, YT, Q-HS, JD, YC, ZZ, Acquisition of data, Analysis and interpretation of data; YT, Acquisition of data, Analysis and interpretation of data, Contributed unpublished essential data or reagents; T-TY, Analysis and interpretation of data, Drafting or revising the article; YEZ, Conception and design, Analysis and interpretation of data, Drafting or revising the article, Contributed unpublished essential data or reagents; SYC, Conception and design, Analysis and interpretation of data, Drafting or revising the article

## Ethics

Animal experimentation: All mice were maintained and handled according to protocols (ASP 13-214) approved by the Animal Care and Use Committee of the National Cancer Institute, NIH.

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
