## [Decision Letter]

Thank you for sending your work entitled “Requirement of Smurf-mediated
endocytosis of Patched1 in Sonic Hedgehog signal reception” for consideration
at *eLife.* Your article has been favorably evaluated by a Senior
editor and 3 reviewers, one of whom is a member of our Board of Reviewing
Editors.

The following individuals responsible for the peer review of your submission have
agreed to reveal their identity: Robb Krumlauf (Reviewing editor); Jin Jiang and Ben
Allen (peer reviewers).

The Reviewing editor (Robb Krumlauf) and the other reviewers (Jin Jiang, Ben Allen,
and a third anonymous reviewer) discussed their comments before we reached this
decision, and the Reviewing editor has assembled the following comments to help you
prepare a revised submission.

The consensus view of all of the reviewers is that the work is potentially of
significant interest and could represent an important advance in the field. However,
each reviewer has substantial concerns that would need to be addressed before
publication of the paper could be considered. This involves additional
experimentation and major revisions to the text. There are also issues raised over
interpretation of data and missing key citations. Under normal circumstances
requests for such substantial revisions would lead to a decision to reject the
paper, but in this case because the reviewers would like to see the paper published
if their concerns are met we wish to offer the opportunity for a revision.

To aid the revision process in this case we provide the specific comments of all
three reviewers.

*Reviewer #1*:

In this manuscript, Yue et al investigated the role of Smurf-mediated Ptch1
ubiquitination in the regulation of Shh signaling. They provided evidence that Shh
promotes Ptch1 enrichment in the lipid rafts and that the PPXY sorting signals (PY
motifs) in Ptch1 promotes endocytosis and degradation of Ptch1. They further showed
that the PY motifs are required for Shh-induced ciliary exit of Ptch1 and optimal Hh
pathway activation. They identified Smurf 1 and 2 as two E3 critical ligases that
promote Ptch1 ubiquitination and degradation through the PY motifs. Interestingly,
they found that the expression of Smurf1/2 is upregulated in response to Shh. By
using FRET and CoIP, they provided evidence that Smurf and Ptch1 physically interact
depending on the PY motifs. Finally, they showed that genetic ablation of Smurf1 and
2 specifically affected Shh-induced proliferation of GCPs. Overall, the experiments
were well executed and the data are convincing. The work is complementary to a
recent publication that mainly described a role of Smurf in targeting Drosophila Ptc
(Huang et al., PLOS Bio 2013), and represents an important advance in the field,
allowing one to compare and contrast the fly and mammalian systems. However, the
authors should address the following concerns either by discussion or by additional
experiments before publication is recommended.

1) Whereas the evidence for Smurf/PY-mediated Ptch1 endocytosis and degradation is
strong, it is not so clear how this process is promoted by Shh. Although Shh-induced
upregulation of Smurf could contribute, other mechanisms may exist. Have the authors
examined whether Shh promotes the binding of Smurf to Ptch1? For example, does Shh
treatment increase the FRET between GFP-Smurf2CG and Ptch1-RFP shown in Figure 9? On the other hand, Smurf-mediated
degradation of Ptch1 could be Shh independent, as suggested by Casali (Science
Signaling, 2010). For example, Ptch1Δ2, which has the Shh binding domain
deleted (Briscoe et al., Mol Cell 2001), might still be regulated by Smurf/PY.
Furthermore, Huang et al suggested that Smurf prefers degrading ligand-unbound Ptc
(Huang et al., PLOS Bio 2013). How could the authors reconcile their finding that
Shh promotes Ptch1 degradation? Could they examine whether Ptch1Δ2 is
degraded more or less effectively by Smurf than Ptch1 in the presence of Shh?

2) Huang et al argued that Smurf-mediated ubiquitination and degradation of Ptc are
promoted by activated forms of Smo (Smo^SD^) in Drosophila (Huang et al.,
PLOS Bio 2013). Have the authors examined whether Shh promotes Ptch1 degradation
through Smo? For example, does overexpression of mammalian Smo^SD^ promote
Smurf-mediated ubiquitination/degradation of Ptch1?

3) The authors showed that mutating the PY motifs or Smurf1/2 affected both Ptch1
ciliary exit and endocytosis. Is the failure of Ptch1 ciliary exit the result of
defective endocytosis? Or could Shh induce Ptch1 ubiquitination in the primary
cilium, which may directly regulate ciliary exit of Ptch1? Is there any evidence
that Smurf1/2 can be found in the primary cilium with or without Shh treatment? In
Figure 6, can Shh trigger ciliary exit
of Ptch1 in the absence of Smurf1/2? Does pharmacological blockage of Ptch1
endocytosis/degradation affect Ptch1 ciliary exit?

4) The effect of Δ2PY-GFP on Smo ciliary localization presented in Figure 4 does not match the quantification
well, especially at 4 hours after Shh treatment where there is almost no difference
in the ciliary Smo levels between Δ2PY-GFP and Ptch1-GFP (Figure 4) while there is a 2-told difference
in the quantification (Figure 4). The
authors need to provide a better image reflecting the quantification. Of note, it
has been shown that Shh/Ptch1 regulates both the ciliary localization and
conformation of Smo (Zhao et al., nature 2007). Have the authors examined whether
Δ2PY-GFP affect mSmo conformation using FRET analysis?

*Reviewer #2*:

In this manuscript Yue et al. uncover a role for the Hect E3 ligases Smurf1 and
Smurf2 in promoting Hedgehog-dependent changes in the subcellular localization of
the Patched receptor that leads to their increased turnover. This Smurf1/2-mediated
trafficking of Patched is also important for its exit from primary cilia during
pathway activation. Using cultured cells including MEFs knockout for Smurf1&2,
the authors show that this trafficking event is, in turn, important for the ciliary
accumulation of Smoothened and for the activation of Gli-mediated transcription. The
authors show that the Shh-promoted proliferation of granule cell progenitors
requires Smurf1 and Smurf2, suggesting an important function of this regulatory
mechanism in a well-characterized physiological context dependent on Hedgehog
ligands.

This is an interesting manuscript that adds to our understanding of the molecular
mechanisms underlying Hedgehog signaling. In particular, although it has been
speculated that endocytic trafficking may be implicated in Patched and Smoothened
localization, ciliary accumulation and signaling, the molecular mechanisms
implicated in this process are poorly defined.

The strongest aspects of this manuscript are the loss of function experiments
conducted with the Sf1-/-,Sf2fl/fl MEFs and GPCs. Indeed, the complete absence of
Sf1 and Sf2 leads to a remarkable inhibition of Smo and Gli3 ciliary localization
and blunting of Shh-promoted induction of Gli1 levels in MEFs. These results are
strongly supported by the experiments in Figure 11 showing a reduction of neural progenitors in cerebellar slice cultures
knockout for Sf1 and Sf2 and an inability of Shh to promote the in vitro
proliferation of granule cell progenitors when Sf1 and Sf2 are knocked out. These
experiments strongly support an important functional requirement of Smurf proteins
for Hedgehog signal transduction.

In terms of mechanisms describing the function of Smurf proteins, the evidence
presented in the manuscript are however disappointing in that they are too often not
convincing, confusing or incomplete. For example, according to their model, Hedgehog
ligands are shown to promote the localization of Patched in caveolae, a transitory
localization that promote the Smurf dependent ubiquitination of Patched and its
endosomal routing to the lysosomes where it is degraded. First of all, although
scattered evidence suggests that caveosomes and endosomes may physically interact in
specific contexts, the authors present their evidence supporting a role of Rab
proteins and endosomal trafficking in promoting Patched exit from caveolae as a well
defined and accepted mechanism. However, caveolae-mediated endocytosis is most often
described to be separate from endosomal sorting. Although this could represent a
novel sorting mechanism for cell surface receptors, the characterization of this
process needs to be strengthened and better discussed.

Moreover, all of the evidence supporting the localization of Ptch in different
subcellular fraction relies on overexpression experiments and on colocalization with
overexpressed markers tagged with fluorescent proteins (especially important for
Rab7). These experiments should be repeated using endogenous proteins and images
obtained at higher resolution to more precisely follow the fate of Ptch trafficking
and more convincingly support the implication of caveolae and/or endosomal
trafficking.

In addition in my opinion the biggest question that is left unanswered is how
ubiquitination of Patched by Smurf proteins contributes to its function. Do Smurfs
lead to Patched mono-ubiquitination or to K63 or K48 ubiquitin chain conjugations?
Is ubiquitination involved in Patched endocytosis per se or in its sorting from
endosomes to lysosomes? Does Hedgehog ligand promote the interaction of Patched with
Smurfs? Do Hedgehog ligands promote Patched ubiquitination?

There also seems to be a disconnection between the results obtained using the
Ptch-d2PY mutant (when rescuing the Ptch1-/- MEFs) and the results obtained in the
Sf1, Sf2 double KO cells. Indeed, whereas the Shh-promoted accumulation of Smo and
Gli1 activation are blunted in the dKO cells, Smo accumulation is only reduced when
the d2PY mutant is expressed (4C,D). Since the interaction between the d2PY mutant
and Smurf proteins seems to be completely abolished (9E) how is this explained? If
there is more Ptch1-d2PY in cilia, why do Smo enters at all?

*Reviewer #3*:

In the manuscript entitled “Requirement of Smurf-Mediated Endocytosis of
Patched 1 in Sonic Hedgehog Signal Reception”, Yu et al. present evidence
that Smurf1 and Smurf2 promote ubiquitination of PTCH1 resulting in endocytic
turnover that is required for HH pathway activation. In particular, the authors
provide significant experimental data examining the subcellular localization of
PTCH1 and the role of two PPXY motifs in regulating PTCH1 localization turnover, and
downstream effects on HH pathway function. While, overall the results appear to be
of high quality, there are some issues with both interpretation of the data and
proper acknowledgement of previous work that the authors must address.

Major comments:

1) There is an unfortunate lack of proper citation of previous work by other labs in
this field. Two essential examples include the recent publication of work
identifying a role for Smurfs in regulating Drosophila Ptc turnover (Huang et al.,
PLOS Biology, 2013), and work from Tom Kornberg that defined a role for the PPXY
motif in regulating the turnover of vertebrate PTCH1 (Kawamura et al., JBC, 2008).
These two papers directly impact the current study by Yue et al., and this work
should be considered in the context of these previous studies.

2) In Figure 5, the authors utilize Ptch1-/-
MEFs to address differences in the ability of PTCH1 and PTCH1Δ2PY to promote
ligand-dependent signaling. However, the authors miss an opportunity to distinguish
between the ligand-dependent and ligand-independent effects of PTCH1 in the HH
pathway. They should use these cells and constructs to examine the ability of PTCH1
or PTCH1Δ2PY to antagonize SMO in the absence of ligand. That is, Ptch1-/-
MEFs display constitutive HH pathway activation; however, re-expressing PTCH1
rescues this pathway activity. The question is whether PTCH1Δ2PY is equally
effective? Do the authors observe equivalent antagonism of SMO in these cells? Or is
PTCH1Δ2PY a more effective antagonist of SMO than wt PTCH1? These are
straightforward questions to address since the authors have all the necessary tools
and reagents in hand.

---

## [Author Response]

Reviewer #1:

*In this manuscript, Yue et al investigated the role of Smurf-mediated Ptch1
ubiquitination in the regulation of Shh signaling. […] However, the
authors should address the following concerns either by discussion or by
additional experiments before publication is recommended*.

*1) Whereas the evidence for Smurf/PY-mediated Ptch1 endocytosis and
degradation is strong, it is not so clear how this process is promoted by Shh.
Although Shh-induced upregulation of Smurf could contribute, other mechanisms
may exist. Have the authors examined whether Shh promotes the binding of Smurf
to Ptch1? For example, does Shh treatment increase the FRET between GFP-Smurf2CG
and Ptch1-RFP shown in*
Figure 9*? On the
other hand, Smurf-mediated degradation of Ptch1 could be Shh independent, as
suggested by Casali (Science Signaling, 2010). For example, Ptch1Δ2,
which has the Shh binding domain deleted (Briscoe et al., Mol Cell 2001), might
still be regulated by Smurf/PY. Furthermore, Huang et al suggested that Smurf
prefers degrading ligand-unbound Ptc (Huang et al., PLOS Bio 2013). How could
the authors reconcile their finding that Shh promotes Ptch1 degradation? Could
they examine whether Ptch1Δ2 is degraded more or less effectively by
Smurf than Ptch1 in the presence of Shh?*

We thank this reviewer for raising these very important issues. Our previous and new
data indicate that Shh promotes the Smurf-mediated endocytosis of Ptch1 in several
ways. First, Smurfs are preferentially localized in the nucleus in normal cells
(Kavsak et al., Mol Cell 6:1365-75, 2000) and play important roles in maintaining
the genomic stability (Blank et al, Nature Medicine 18:227-34, 2012). In the revised
manuscript, we show that Shh promotes a re-pivoting of Smurf2 from the nucleus to
the cytoplasm (Figure 7, and Figure 7—figure supplement 1). Second,
our data also show that Shh induces Smurfs expression (Figure 5). So, these two events should lead to an
increase of the effective cytoplasmic concentration of Smurfs. Third, as requested,
we conducted a new FRET experiment and found that Shh indeed promotes the
colocalization of Ptch1 and Smurf2 (Figure 7). Fourth, we further add new data showing that ShhN treatment enhances the
ubiquitin modification of Ptch1 (Figure 9),
consistent with our data showing that Shh promotes Ptch1 turnover (Figure 8).

In Huang et al, the authors ectopically expressed activated Smo mutants,
Smo^SD^, in the entire A-compartment, which drastically increased the
level of Ptc (Huang et al, Figure 4). They
argue that DSmurf prefers the ligand-unbound Ptc as a substrate because ectopic
expression of DSmurf reduced Ptc staining selectively in the A compartment. However,
comparing their Figure 4, one could
find that the intensity of Ptc staining at the A/P boundary was also reduced by
DSmurf, notwithstanding the fact that Ptc is normally high at the boundary. On the
other hand, since the authors did not examine the distribution of Hh in the disc
that received the ectopically expressed Smo^SD^, it would be an unsupported
assumption that the elevated Ptc in the A compartment was still in the unbound form.
After all, the Hh ligand is normally restricted to the compartmental border by the
high level of Ptc there. If the border stripe of high level Ptc was made to expand,
Hh zone should expand with it. Furthermore, it is well known in the field that Ptc
and Smo, when over-expressed, tend to form a nonphysiological complex (Stone et al,
Nature 384:129, 1996, and Taipale et al, Nature 418:892, 2002).This raises a
possibility that the nonphysiological Ptc-Smo complex could trigger an
“unfolded protein response” of some sort that leads to the
DSmurf-mediated destruction. This type of degradation is very different from the one
that we describe in our manuscript, although both could be mediated by the Smurf E3
ubiquitin ligases, even specifically.

Notwithstanding the above analysis, assuming DSmurf does prefer the ligand unbound
form of Ptc for degradation, this would put the site of DSmurf action in the A
compartment, where Ptc level is low and Smo is in an inactive state. However, their
data indicated that Smo has to be activated in order to promote Ptc degradation. In
Huang et al, there is no data that either indicate or imply the source of the
activated Smo for activating the Smurf-mediated Ptc turnover or to explain this
conspicuous conflict.

We measured the turnover rate of the loop2 mutant of Ptch1 in wt MEFs, and found that
the effect of Shh ligand induction was abolished (Figure 8). We further quantified the turnover rate of Ptch1 in
Smo^null^ cells, and found that Shh still promotes Ptch1 turnover there
(Figure 8). Moreover, we did not detect
interaction between Smo and Smurfs by co-IP experiments, even though Smurf was shown
to bind Ptch1 readily (Figure 7—figure supplement 2). So, these results demonstrate that Smurfs likely promote
Ptch1 endocytic turnover through direct binding, rather than using Smo as an
intermediate, as suggested by Huang et al. However, Smo probably still has a long
term feedback role through enhancing downstream Smurf gene expression.

*2) Huang et al argued that Smurf-mediated ubiquitination and degradation of
Ptc are promoted by activated forms of Smo
(Smo*^*SD*^*) in Drosophila (Huang et
al., PLOS Bio 2013). Have the authors examined whether Shh promotes Ptch1
degradation through Smo? For example, does overexpression of mammalian
Smo*^*SD*^
*promote Smurf-mediated ubiquitination/degradation of Ptch1?*

As stated above, we examined Ptch1 turnover in Smo^null^ cells, and found
that Shh still promotes Ptch1 turnover. We also found by Co-IP experiment that Ptch1
binds Smurf but Smo does not (Figure 7—figure supplement 2). These data strongly argue that Shh-induced,
Smurfs-mediated Ptch1 endocytic turnover is independent of Smo.

*3) The authors showed that mutating the PY motifs or Smurf1/2 affected both
Ptch1 ciliary exit and endocytosis. Is the failure of Ptch1 ciliary exit the
result of defective endocytosis? Or could Shh induce Ptch1 ubiquitination in the
primary cilium, which may directly regulate ciliary exit of Ptch1? Is there any
evidence that Smurf1/2 can be found in the primary cilium with or without Shh
treatment? In*
Figure 6*, can Shh
trigger ciliary exit of Ptch1 in the absence of Smurf1/2? Does pharmacological
blockage of Ptch1 endocytosis/degradation affect Ptch1 ciliary
exit?*

It is our interpretation that Ptch1Δ2PY fails to exit cilia because of
defective endocytosis. Despite an initial hypothesis, we found neither endogenous
nor transfected Smurfs in the cilia with or without Shh treatment. Our data also
show that the Shh-induced ciliary export of Ptch1 was compromised when Smurf1 and
Smurf2 were knocked down with siRNAs (Figure 5—figure supplement 1). We further show that blocking Ptch1
endocytosis with Leupeptin also blocked its ciliary exit (this result were not
included in the previous submission, but is now added as Figure 3—figure supplement 1 in the revised
manuscript).

*4) The effect of Δ2PY-GFP on Smo ciliary localization presented
in*
Figure 4
*does not match the quantification well, especially at 4 hours after Shh
treatment where there is almost no difference in the ciliary Smo levels between
Δ2PY-GFP and Ptch1-GFP (*Figure 4*) while there is a 2-told
difference in the quantification (*Figure 4*). The authors need to provide a
better image reflecting the quantification. Of note, it has been shown that
Shh/Ptch1 regulates both the ciliary localization and conformation of Smo (Zhao
et al., nature 2007). Have the authors examined whether Δ2PY-GFP affect
mSmo conformation using FRET analysis?*

We replaced the images in the old Figure 4
with better ones in the revision (new Figure 3). By using a sophisticated FRET imaging approach, Zhao et al elegantly
demonstrated that Hh induces phosphorylation and a conformational change of Smo
c-tail that result in Smo dimerization and activation of downstream signaling. Their
work also extended this observation to mammalian Smo. However, this regulation,
albeit a likely key event in the Shh pathway activation, lies downstream to Ptch1
functions. Since we have demonstrated that Shh-induced Ptch1 endocytic turnover is
independent of Smo, and analyzed extensively the ciliary trafficking of Smo, another
well recognized key event of the Shh pathway activation, we felt that examining
Δ2PY-GFP on mSmo conformation would be a repetition of an already
well-addressed issue. In addition, setting up the FRET experiment on Smo
conformation would not be a trivial endeavor, if one needs to do it properly. If
this reviewer and the editors deem this FRET experiment absolutely essential, which
we would respectfully disagree, we will perform as demanded, provided that we are
granted additional time.

Reviewer #2:

*In this manuscript Yue et al. uncover a role for the Hect E3 ligases Smurf1
and Smurf2 in promoting Hedgehog-dependent changes in the subcellular
localization of the Patched receptor that leads to their increased
turnover*. *[…]*

*The strongest aspects of this manuscript are the loss of function experiments
conducted with the Sf1-/-,Sf2fl/fl MEFs and GPCs. Indeed, the complete absence
of Sf1 and Sf2 leads to a remarkable inhibition of Smo and Gli3 ciliary
localization and blunting of Shh-promoted induction of Gli1 levels in MEFs.
These results are strongly supported by the experiments in*
Figure 11
*showing a reduction of neural progenitors in cerebellar slice cultures
knockout for Sf1 and Sf2 and an inability of Shh to promote the in vitro
proliferation of granule cell progenitors when Sf1 and Sf2 are knocked out.
These experiments strongly support an important functional requirement of Smurf
proteins for Hedgehog signal transduction*.

*In terms of mechanisms describing the function of Smurf proteins, the
evidence presented in the manuscript are however disappointing in that they are
too often not convincing, confusing or incomplete. For example, according to
their model, Hedgehog ligands are shown to promote the localization of Patched
in caveolae, a transitory localization that promote the Smurf dependent
ubiquitination of Patched and its endosomal routing to the lysosomes where it is
degraded. First of all, although scattered evidence suggests that caveosomes and
endosomes may physically interact in specific contexts, the authors present
their evidence supporting a role of Rab proteins and endosomal trafficking in
promoting Patched exit from caveolae as a well defined and accepted mechanism.
However, caveolae-mediated endocytosis is most often described to be separate
from endosomal sorting. Although this could represent a novel sorting mechanism
for cell surface receptors, the characterization of this process needs to be
strengthened and better discussed*.

We agree with this reviewer that caveolae was a recently recognized alternative route
for internalization of membrane-bound ligand-receptor complexes, but this phenomenon
was actually noted more than two decades ago. At that time, a term of
“potocytosis” was coined to distinguish it from the Clathrin-mediated
endocytosis (Anderson RG, Science 255:410-1, 1992; Gleizes PE, Eur. J. Cell Biology
71:144-53, 1996), because the cargo of potocytosis was thought to be emptied
directly into the cytosol. Later studies demonstrated that caveolae-mediated
internalization actually feeds into the conventional endocytic pathway, and
“caveosomes”, which were previously regarded as independent organelles
distinct from endosomes, were actually late endosomes modified by the accumulated
Caveolin-1 therein (Hayer et al, J Cell Biol 191:615-29, 2010; Sandvig et al, Curr
Opin Cell Biol 23:413-420, 2011). To clarify this issue, we made modifications in
the Introduction and cited several key references.

*Moreover, all of the evidence supporting the localization of Ptch in
different subcellular fraction relies on overexpression experiments and on
colocalization with overexpressed markers tagged with fluorescent proteins
(especially important for Rab7). These experiments should be repeated using
endogenous proteins and images obtained at higher resolution to more precisely
follow the fate of Ptch trafficking and more convincingly support the
implication of caveolae and/or endosomal trafficking*.

Antibodies again mouse Ptch1 are not commercially available, precluding a direct
visualization of the endogenous Ptch1, which is present at extremely low level in
cells (Rohatgi et al Science). Fluorescence labeled Rab5, Rab7, and Lamp1 are widely
used for marking early endosomes, late endosomes, and lysosomes, and the data in
question were generated through confocal imaging on a newly acquired Zeiss LSM710
microscope. We have repeated the experiments in question using Ptch1GFP and
antibodies against endogenous Rab5, Rab7, and Lamp1, respectively. The data are
displayed in new Figure 2 and Figure 2—figure supplement 1 and Figure 2—figure supplement 3. Signals from antibody staining of endogenous proteins were quite low,
probably reflecting the low abundance of the interacting species or the low avidity
of this commercial antibody, nevertheless, colocalization between Ptch1GFP and Rab7
poitive late endosomes was confirmed. We also showed colocalization between Ptch1GFP
and Lamp1 positive lysosomes using leupeptin to block proteolysis. However, we were
unable to detect colocalization between Ptch1GFP and early endosomes (Rab5) without
or with ShhN treatment, confirming our previous finding that Ptch1 traverses from
lipid rafts directly to late endosomes, bypassing early endosomes. Finally, the d2PY
mutant was not colocalized with any of these vesicles. We want to emphasize that
these confocal images presented were taken in z-stack using a 63x oil lens. Some
images may appear fuzzy, particularly in colocalizing areas/vesicles. This is likely
because only a very small fraction of cytoplasmic Ptch1 is channeled to the
endocytic pathway; the bulk of forced expressed Ptch1 still turns over via
proteasomes (Figure 1).

In addition in my opinion the biggest question that is left unanswered is how
ubiquitination of Patched by Smurf proteins contributes to its function. Do
Smurfs lead to Patched mono-ubiquitination or to K63 or K48 ubiquitin chain
conjugations? Is ubiquitination involved in Patched endocytosis per se or in its
sorting from endosomes to lysosomes? Does Hedgehog ligand promote the
interaction of Patched with Smurfs? Do Hedgehog ligands promote Patched
ubiquitination?

In our humble opinion, elucidation of the type of Smurfs-mediated ubiquitin
modification of Ptch1is certainly informative, but is nevertheless a mechanistic
detail in our investigation. It is also extremely difficult to visualize
monoubiquitination of Ptch1 under natural settings, given the size of this protein.
We did however use mutant forms of ubiquitin and found that Smurf2 promotes Ptch1 to
undergo both K63 and K48 ubiquitin chain-mediated ubiquitination (new Figure 9). We further show that Shh-N promotes
interaction of Ptch1 with Smurfs (new Figure 7) and Ptch1 polyubiquitination (new Figure 9). Because Ptch1 ΔPY is accumulated in Caveolin-positive
lipid raft but not in late endosome (Figures 1 and 2), we believe that Smurf-mediated Ptch1 ubiquitination is
involved in sorting of Ptch1 from lipid raft to late endosomes.

There also seems to be a disconnection between the results obtained using the
Ptch-d2PY mutant (when rescuing the Ptch1-/- MEFs) and the results obtained in
the Sf1, Sf2 double KO cells. Indeed, whereas the Shh-promoted accumulation of
Smo and Gli1 activation are blunted in the dKO cells, Smo accumulation is only
reduced when the d2PY mutant is expressed (4C,D). Since the interaction between
the d2PY mutant and Smurf proteins seems to be completely abolished (9E) how is
this explained? If there is more Ptch1-d2PY in cilia, why do Smo enters at
all?

We replaced the d2PY images in old Figure 4
as well as those in old Figure 8 with new
ones that better reflect the corresponding statistic graphs. We apologize for those
images that may have exaggerated the difference. Judging from the data graphs, it is
clear that the reduction in Smo ciliary localization and Gli1 activation caused by
d2PY deletion is clearly in line with that by Smurfs knockdown (compare Figure 3, time point 1-4 hours vs. Figure 6).

With regard to the last question, the current paradigm of Ptch1 inhibiting Smo by
preventing the latter entry into cilia is based on the observation that Smo moves in
whereas Ptch1 moves out of cilia under the influence of Shh (Rohatgi et al, Science
317:372-8, 2007). However, there is no evidence to indicate that the presence of
these two membrane receptors in the cilium is mutually exclusive. To the contrary,
there are published studies reporting cyclopamine actually promotes Smo entry into
the cilium, suggesting that Smo and Ptch1 can co-exist in cilia.

Reviewer #3:

*In the manuscript entitled “Requirement of Smurf-Mediated Endocytosis
of Patched 1 in Sonic Hedgehog Signal Reception”, Yu et al. present
evidence that Smurf1 and Smurf2 promote ubiquitination of PTCH1 resulting in
endocytic turnover that is required for HH pathway activation. In particular,
the authors provide significant experimental data examining the subcellular
localization of PTCH1 and the role of two PPXY motifs in regulating PTCH1
localization turnover, and downstream effects on HH pathway function. While,
overall the results appear to be of high quality, there are some issues with
both interpretation of the data and proper acknowledgement of previous work that
the authors must address*.

Major comments:

*1) There is an unfortunate lack of proper citation of previous work by other
labs in this field. Two essential examples include the recent publication of
work identifying a role for Smurfs in regulating Drosophila Ptc turnover (Huang
et al., PLOS Biology, 2013), and work from Tom Kornberg that defined a role for
the PPXY motif in regulating the turnover of vertebrate PTCH1 (Kawamura et al.,
JBC, 2008). These two papers directly impact the current study by Yue et al.,
and this work should be considered in the context of these previous
studies*.

We have cited these two papers and discussed extensively the Huang’s recent
publication.

*2) In*
Figure 5*, the
authors utilize Ptch1-/- MEFs to address differences in the ability of PTCH1 and
PTCH1Δ2PY to promote ligand-dependent signaling. However, the authors
miss an opportunity to distinguish between the ligand-dependent and
ligand-independent effects of PTCH1 in the HH pathway. They should use these
cells and constructs to examine the ability of PTCH1 or PTCH1Δ2PY to
antagonize SMO in the absence of ligand. That is, Ptch1-/- MEFs display
constitutive HH pathway activation; however, re-expressing PTCH1 rescues this
pathway activity. The question is whether PTCH1Δ2PY is equally effective?
Do the authors observe equivalent antagonism of SMO in these cells? Or is
PTCH1Δ2PY a more effective antagonist of SMO than wt PTCH1? These are
straightforward questions to address since the authors have all the necessary
tools and reagents in hand*.

We did the experiment as requested and the results indicate that Δ2PY is
equally effective as the wt Ptch1 in antagonizing Smo in Ptch1-/- MEFs (Figure 4). This is different from the results
obtained from Shh-induced signaling events.